# Neural Oscillators are Universal

**Samuel Lanthaler**
California Institute of Technology
slanth@caltech.edu

**T. Konstantin Rusch**
ETH Zurich

**Siddhartha Mishra**
ETH Zurich

## Abstract

Coupled oscillators are being increasingly used as the basis of machine learning (ML) architectures, for instance in sequence modeling, graph representation learning and in physical neural networks that are used in analog ML devices. We introduce an abstract class of *neural oscillators* that encompasses these architectures and prove that neural oscillators are universal, i.e, they can approximate any continuous and casual operator mapping between time-varying functions, to desired accuracy. This universality result provides theoretical justification for the use of oscillator based ML systems. The proof builds on a fundamental result of independent interest, which shows that a combination of forced harmonic oscillators with a nonlinear read-out suffices to approximate the underlying operators.

## 1  Introduction

Oscillators are ubiquitous in the sciences and engineering [12, 30]. Prototypical examples include pendulums in mechanics, feedback and relaxation oscillators in electronics, business cycles in economics and heart beat and circadian rhythms in biology. Particularly relevant to our context is the fact that the neurons in our brain can be thought of as oscillators on account of the periodic spiking and firing of the action potential [29, 11]. Consequently, functional brain circuits such as cortical columns are being increasingly analyzed in terms of networks of coupled oscillators [29].

Given this wide prevalence of (networks of) oscillators in nature and man-made devices, it is not surprising that oscillators have inspired various machine learning architectures in recent years. Prominent examples include the *CoRNN* [27] and *UnICORNN* [28] recurrent neural networks for sequence modeling. CoRNN is based on a network of coupled, forced and damped oscillators, whereas UnICORNN is a multi-layer sequence model that stacks networks of independent undamped oscillators as hidden layers within an RNN. Both these architectures were rigorously shown to mitigate the exploding and vanishing gradient problem [20] that plagues RNNs. Hence, both CoRNN and UnICORNN performed very well on sequence learning tasks with *long-term dependencies*. Another example of the use of oscillators in machine learning is provided by *GraphCON* [26], a framework for designing graph neural networks (GNNs) [3], that is based on coupled oscillators. GraphCON was also shown to ameliorate the *oversmoothing* problem [25] and allow for the deployment of multi-layer deep GNNs. Other examples include Second Order Neural ODEs (SONODEs) [19], which can be interpreted as oscillatory neural ODEs, locally coupled oscillatory recurrent networks (LocoRNN) [17], and Oscillatory Fourier Neural Network (O-FNN) [13].

Another avenue where ML models based on oscillators arise is that of *physical neural networks* (PNNs) [35] i.e., physical devices that perform machine learning on analog (beyond digital) systems. Such analog systems have been proposed as alternatives or accelerators to the current paradigm of machine learning on conventional electronics, allowing us to significantly reduce the prohibitive energy costs of training state-of-the-art ML models. In [35], the authors propose a variety of physical neural networks which include a mechanical network of multi-mode oscillations on a plate and electronic circuits of oscillators as well as a network of nonlinear oscillators. Coupled with a novel *physics aware training* (PAT) algorithm, the authors of [35] demonstrated that their

37th Conference on Neural Information Processing Systems (NeurIPS 2023).

nonlinear oscillatory PNN achieved very good performance on challenging benchmarks such as Fashion-MNIST [36]. Moreover, other oscillatory systems such as coupled lasers and spintronic nano-oscillators have also been proposed as possible PNNs, see [33] as an example of the use of thermally coupled vanadium dioxide oscillators for image recognition and [24, 32] for the use of spin-torque nano-oscillators for speech recognition and for neuromorphic computing, respectively.

What is the rationale behind the successful use of (networks of) oscillators in many different contexts in machine learning? The authors of [27] attribute it to the inherent stability of oscillatory dynamics, as the state (and its gradients) of an oscillatory system remain within reasonable bounds throughout the time-evolution of the system. However, this is at best a partial explanation, as it does not demonstrate why oscillatory dynamics can learn (approximate) mappings between inputs and outputs rather than bias the learned states towards oscillatory functions. As an example, consider the problem of classification of MNIST [18] (or Fashion-MNIST) images. It is completely unclear if the inputs (vectors of pixel values), outputs (class probabilities) and the underlying mapping possess any (periodic) oscillatory structure. Consequently, how can oscillatory RNNs (such an CoRNN and UnICORNN) or a network of oscillatory PNNs learn the underlying mapping?

Our main aim in this paper is to provide an answer to this very question on the ability of neural networks, based on oscillators, to express (to approximate) arbitrary mappings. To this end,

- We introduce an abstract framework of *neural oscillators* that encompasses both sequence models such as CoRNN and UnICORNN, as well as variants of physical neural networks as the ones proposed in [35]. These neural oscillators are defined in terms of second-order versions of *neural ODEs* [4], and combine *nonlinear dynamics* with a *linear read-out*.

- We prove a *Universality* theorem for neural oscillators by showing that they can approximate, to any given tolerance, continuous operators between appropriate function spaces.

- Our proof of universality is based on a novel theoretical result of independent interest, termed the *fundamental Lemma*, which implies that a suitable combination of *linear oscillator dynamics* with *nonlinear read-out* suffices for universality.

Such universality results, [1, 5, 14, 21] and references therein, have underpinned the widespread use of traditional neural networks (such as multi-layer perceptrons and convolutional neural networks). Hence, our universality result establishes a firm mathematical foundation for the deployment of neural networks, based on oscillators, in myriad applications. Moreover, our constructive proof provides insight into how networks of oscillators can approximate a large class of mappings.

## 2   Neural Oscillators

**General Form of Neural Oscillators.**   Given $u : [0, T] \to \mathbb{R}^p$ as an input signal, for any final time $T \in \mathbb{R}_+$, we consider the following system of *neural* ODEs for the evolution of dynamic hidden variables $y \in \mathbb{R}^m$, coupled to a linear read-out to yield the output $z \in \mathbb{R}^q$,

$$
\begin{cases}
\ddot{y}(t) = \sigma\left(Wy(t) + Vu(t) + b\right), & \text{(2.1a)} \\
y(0) = \dot{y}(0) = 0, & \text{(2.1b)} \\
z(t) = Ay(t) + c. & \text{(2.1c)}
\end{cases}
$$

Equation (2.1) defines an input-/output-mapping $u(t) \mapsto z(t)$, with time-dependent output $z : [0, T] \to \mathbb{R}^q$. Specification of this system requires a choice of the hidden variable dimension $m$ and the activation function $\sigma$. The resulting mapping $u \mapsto z$ depends on tunable weight matrices $W \in \mathbb{R}^{m \times m}$, $V \in \mathbb{R}^{m \times p}$, $A \in \mathbb{R}^{q \times m}$ and bias vectors $b \in \mathbb{R}^m$, $c \in \mathbb{R}^q$. For simplicity of the exposition, we consider only *activation functions* $\sigma \in C^\infty(\mathbb{R})$, *with* $\sigma(0) = 0$ *and* $\sigma'(0) = 1$, such as $\tanh$ or $\sin$, although more general activation functions can be readily considered. This general second-order neural ODE system (2.1) will be referred to as a **neural oscillator**.

**Multi-layer neural oscillators.** As a special case of neural oscillators, we consider the following *much sparser* class of second-order neural ODEs,

$$
\begin{cases}
y^0(t) := u(t), & \text{(2.2a)} \\
\ddot{y}^\ell(t) = \sigma\left(w^\ell \odot y^\ell(t) + V^\ell y^{\ell-1}(t) + b^\ell\right), & (\ell = 1, \ldots, L), & \text{(2.2b)} \\
y^\ell(0) = \dot{y}^\ell(0) = 0, & \text{(2.2c)} \\
z(t) = Ay^L(t) + c. & \text{(2.2d)}
\end{cases}
$$

In contrast to the general neural oscillator (2.1), the above multi-layer neural oscillator (2.2) defines a *hierarchical* structure; The solution $y^\ell \in \mathbb{R}^{m_\ell}$ at level $\ell$ solves a second-order ODE with driving force $y^{\ell-1}$, and the lowest level, $y^0 = u$, is the input signal. Here, the layer dimensions $m_1, \ldots, m_L$ can vary across layers, the weights $w^\ell \in \mathbb{R}^{m_\ell}$ are given by vectors, with $\odot$ componentwise multiplication, $V^\ell \in \mathbb{R}^{m_\ell \times m_{\ell-1}}$ is a weight matrix, and $b^\ell \in \mathbb{R}^{m_\ell}$ the bias. Given the result of the final layer, $y^L$, the output signal is finally obtained by an affine output layer $z(t) = Ay^L(t) + c$. In the multi-layer neural oscillator, the matrices $V^\ell$, $A$ and vectors $w^\ell$, $b^\ell$ and $c$ represent the trainable hidden parameters. The system (2.2) is a special case of (2.1), since it can be written in the form (2.1), with $y := [y^L, y^{L-1}, \ldots, y^1]^T$, $b := [b^L, \ldots, b^1]^T$, and a (upper-diagonal) block-matrix structure for $W$:

$$
W := \begin{bmatrix}
\text{diag}(w^L) & V^L & 0 & \cdots & & 0 \\
0 & \text{diag}(w^{L-1}) & V^{L-1} & \ddots & & \vdots \\
\vdots & \ddots & & \ddots & & 0 \\
0 & \cdots & 0 & \text{diag}(w^2) & V^2 \\
0 & \cdots & 0 & 0 & \text{diag}(w^1)
\end{bmatrix}, \quad V := \begin{bmatrix} 0 \\ \vdots \\ \vdots \\ 0 \\ V^1 \end{bmatrix} \quad \text{(2.3)}
$$

Given the block-diagonal structure of the underlying weight matrices, it is clear that the multi-layer neural oscillator (2.2) is a much sparser representation of the general neural operator (2.1). Moreover, one can observe from the structure of the neural ODE (2.2) that within each layer, the individual neurons are causally *independent* of each other.

Assuming that $w_i^\ell \neq 0$, for all $1 \leq i \leq m_\ell$ and all $1 \leq \ell \leq L$, we further highlight that the multi-layer neural oscillator (2.2) is a Hamiltonian system,

$$
\dot{y}^\ell = \frac{\partial H}{\partial \dot{y}^\ell}, \quad \ddot{y}^\ell = -\frac{\partial H}{\partial y^\ell}, \quad \text{(2.4)}
$$

with the *layer-wise time-dependent Hamiltonian*,

$$
H(y^\ell, \dot{y}^\ell, t) = \frac{1}{2}\|\dot{y}^\ell\|^2 - \sum_{i=1}^{m_\ell} \frac{1}{w_i^\ell}\widehat{\sigma}(w_i^\ell y_i^\ell + (V^\ell y^{\ell-1})_i + b_i^\ell), \quad \text{(2.5)}
$$

with $\widehat{\sigma}$ being the antiderivative of $\sigma$, and $\|\mathbf{x}\|^2 = \langle \mathbf{x}, \mathbf{x} \rangle$ denoting the Euclidean norm of the vector $\mathbf{x} \in \mathbb{R}^m$ and $\langle \cdot, \cdot \rangle$ the corresponding inner product. Hence, any symplectic discretization of the multi-layer neural oscillator (2.2) will result in a fully reversible model, which can first be leveraged in the context of normalizing flows [23], and second leads to a memory-efficient training, as the intermediate states (i.e., $y^\ell(t_0), \dot{y}^\ell(t_0), y_\ell(t_1), \dot{y}_\ell(t_1), \ldots, y_\ell(t_N), \dot{y}_\ell(t_N)$, for some time discretization $t_0, t_1, \ldots, t_N$ of length $N$) do not need to be stored and can be reconstructed during the backward pass. This potentially leads to a drastic memory saving of $\mathcal{O}(N)$ during training.

## 2.1 Examples of Neural Oscillators

**(Forced) harmonic oscillator.** Let $p = m = q = 1$ and we set $W = -\omega^2$, for some $\omega \in \mathbb{R}$, $V = 1, b = 0$ and the activation function to be identity $\sigma(x) = x$. In this case, the neural ODE (2.1) reduces to the ODE modeling the dynamics of a forced simple harmonic oscillator [12] of the form,

$$
\ddot{y} = -\omega^2 y + u, \quad y(0) = \dot{y}(0) = 0. \quad \text{(2.6)}
$$

Here, $y$ is the displacement of the oscillator, $\omega$ the frequency of oscillation and $u$ is a forcing term that forces the motion of the oscillator. Note that (2.6) is also a particular example of the multi-layer oscillator (2.2) with $L = 1$.

This simple example provides justification for our terminology of neural oscillators, as in general, the hidden state $y$ can be thought of as the vector of displacements of $m$-coupled oscillators, which are coupled together through the weight matrix $W$ and are forced through a forcing term $u$, whose effect is modulated via $V$ and a bias term $b$. The nonlinear activation function mediates possible nonlinear feedback to the system on account of large displacements.

**CoRNN.** The Coupled oscillatory RNN (CoRNN) architecture [27] is given by the neural ODE:

$$\ddot{y} = \sigma\left(Wy + \mathcal{W}\dot{y} + Vu + b\right) - \gamma y - \epsilon \dot{y}.$$

We can recover the neural oscillator (2.1) as a special case of CoRNN by setting $\mathcal{W} = \mathbf{0}, \gamma = \epsilon = 0$; thus, a universality theorem for neural oscillators immediately implies a corresponding universality result for the CoRNN architecture.

**UnICORNN.** The Undamped Independent Controlled Oscillatory RNN (UnICORNN) architecture of [28, eqn. 1] recovers the multi-layer neural oscillator (2.2) in the case where the fundamental frequencies of UnICORNN are automatically determined inside the weight matrix $W$ in (2.1).

**Nonlinear oscillatory PNN of [35].** In [35, SM, Sect. 4.A], the authors propose an analog machine learning device that simulates a network of nonlinear oscillators, for instance realized through coupled pendula. The resulting mathematical model is the so-called simplified Frenkel-Kontorova model [2] given by the ODE system,

$$M\ddot{\theta} = -K\sin(\theta) - C\sin(\theta) + f,$$

where $\theta = (\theta_1, \ldots, \theta_N)$ is the vector of angles across all coupled pendula, $M = \mathrm{diag}(\mu^1, \ldots, \mu^N)$ is a diagonal mass matrix, $f$ an external forcing, $K = \mathrm{diag}(k^1, \ldots, k^N)$ the "spring constant" for pendula, given by $k^i = \mu^i g/\ell$ with $\ell$ the pendulum length and $g$ the gravitational acceleration, and where $C = C^T$ is a symmetric matrix, with

$$C_{\ell\ell} = -\sum_{\ell' \neq \ell} C_{\ell\ell'}, \quad \text{so that} \quad [C\sin(\theta)]_\ell = \sum_{\ell' \neq \ell} C_{\ell\ell'}(\sin(\theta_{\ell'}) - \sin(\theta_\ell)), \tag{2.7}$$

which quantifies the coupling between different pendula. We note that this simplified Frenkel-Kontorova system can also model other coupled nonlinear oscillators, such as coupled lasers or spintronic oscillators [35].

We can bring the above system into a more familiar form by introducing the variable $y$ according to the relationship $Py = \theta$ for a matrix $P$. Substitution of this ansatz then yields $MP\ddot{y} = -(K + C)\sin(Py) + f$; choosing $P = M^{-1}(K + C)$, we find

$$\ddot{y} = -\sin(M^{-1}(K + C)y) + f, \tag{2.8}$$

which can be written in the form $\ddot{y} = \sigma(Wy) + f$ for $\sigma = -\sin(\cdot)$ and $W = M^{-1}(K + C)$. We next make a particular choice of $C$, by choosing it in a block-matrix form

$$C := \begin{bmatrix} \gamma^L I & C^L & 0 & \ldots & 0 \\ C^{L,T} & \gamma^{L-1}I & \ddots & & \vdots \\ 0 & \ddots & & \ddots & 0 \\ \vdots & \ldots & C^{3,T} & \gamma^2 I & C^2 \\ 0 & \ldots & 0 & C^{2,T} & \gamma^1 I \end{bmatrix},$$

and with corresponding mass matrix $M$ in block-matrix form $M = \mathrm{diag}(\mu^L I, \mu^{L-1}I, \ldots, \mu^1 I)$, then with $\rho^\ell := \gamma^\ell/\mu^\ell$, we have

$$M^{-1}C := \begin{bmatrix} \rho^L I & C^L/\mu^L & 0 & \ldots & 0 \\ C^{L,T}/\mu^{L-1} & \rho^{L-1}I & \ddots & & \vdots \\ 0 & \ddots & & \ddots & 0 \\ \vdots & & \ddots & \rho^2 I & C^2/\mu^2 \\ 0 & \ldots & 0 & C^{2,T}/\mu^1 & \rho^1 I \end{bmatrix},$$

With the intent of ordering the masses of different layers, such that $\mu^\ell / \mu^{\ell-1} \sim \epsilon$ is small, we now introduce an ordering parameter $\epsilon > 0$. And we consider the following ordering $\gamma^\ell, C^\ell, \mu^\ell \sim \epsilon^\ell$.

Assuming this ordering, it follows that $\rho^\ell, \frac{C^\ell}{\mu^\ell} = O(1)$, and $\frac{C^\ell}{\mu^{\ell-1}} = O(\epsilon)$. This ordering of the masses introduces an effective one-way coupling, making

$$
M^{-1}C = \begin{bmatrix} \rho^L I & V^L & 0 & \cdots & 0 \\ 0 & \rho^{L-1}I & V^{L-1} & \ddots & \vdots \\ \vdots & & \ddots & & 0 \\ 0 & \cdots & 0 & \rho^2 I & V^2 \\ 0 & \cdots & 0 & 0 & \rho^1 I \end{bmatrix} + O(\epsilon),
$$

upper triangular, up to small terms of order $\epsilon$. We note that the diagonal entries $\rho^\ell$ in $M^{-1}C$ are determined by the off-diagonal terms through the identity (2.7). The additional degrees of freedom in the (diagonal) $K$-matrix in (2.8) can be used to tune the diagonal weights of the resulting weight matrix $W = M^{-1}(K + C)$.

Thus, physical systems such as the Frankel-Kontorova system of nonlinear oscillators can be approximated (to leading order) by multi-layer systems of the form

$$
\ddot{y}^\ell = \sigma\left(w^\ell \odot y^\ell + V^\ell y^{\ell-1}\right) + f^\ell, \tag{2.9}
$$

with $f^\ell$ an external forcing, representing a tunable linear transformation of the external input to the system. The only formal difference between (2.9) and (2.2) is (i) the absence of a bias term in (2.9) and (ii) the fact that the external forcing appears outside of the nonlinear activation function $\sigma$ in (2.9). A bias term could readily be introduced by measuring the angles represented by $y^\ell$ in a suitably shifted reference frame; physically, this corresponds to tuning the initial position $y^\ell(0)$ of the pendula, with $y^\ell(0)$ also serving as the reference value. Furthermore, in our proof of universality for (2.2), it makes very little difference whether the external forcing $f$ is applied inside the activation function, as in (2.2b) resp. (2.1a), or outside as in (2.9); indeed, the first layer in our proof of universality will in fact approximate the *linearized dynamics* of (2.2b), i.e. a forced harmonic oscillator (2.6). Consequently, a universality result for the multi-layer neural oscillator (2.2) also implies universality of variants of nonlinear oscillator-based physical neural networks, such as those considered in [35].

## 3 Universality of Neural Oscillators

In this section, we state and sketch the proof for our main result regarding the universality of neural oscillators (2.1) or, more specifically, multi-layer oscillators (2.2). To this end, we start with some mathematical preliminaries to set the stage for the main theorem.

### 3.1 Setting

**Input signal.** We want to approximate operators $\Phi : u \mapsto \Phi(u)$, where $u = u(t)$ is a time-dependent input signal over a time-interval $t \in [0, T]$, and $\Phi(u)(t)$ is a time-dependent output signal. We will assume that the input signal $t \mapsto u(t)$ is continuous, and that $u(0) = 0$. To this end, we introduce the space

$$
C_0([0, T]; \mathbb{R}^p) := \{u : [0, T] \to \mathbb{R}^p \mid t \mapsto u(t) \text{ is continuous and } u(0) = 0\}.
$$

We will assume that the underlying operator defines a mapping $\Phi : C_0([0, T]; \mathbb{R}^p) \to C_0([0, T]; \mathbb{R}^q)$.

The approximation we discuss in this work are based on oscillatory systems starting from rest. These oscillators are forced by the input signal $u$. For such systems the assumption that $u(0) = 0$ is necessary, because the oscillator starting from rest takes a (arbitrarily small) time-interval to synchronize with the input signal (to "warm up"); If $u(0) \neq 0$, then the oscillator cannot accurately approximate the output during this warm-up phase. This intuitive fact is also implicit in our proofs. We will provide a further comment on this issue in Remark 3.2, below.

**Operators of interest.** We consider the approximation of an operator $\Phi : C_0([0, T]; \mathbb{R}^p) \to C_0([0, T]; \mathbb{R}^q)$, mapping a continuous input signal $u(t)$ to a continuous output signal $\Phi(u)(t)$. We will restrict attention to the uniform approximation of $\Phi$ over a compact set of input functions $K \subset C_0([0, T]; \mathbb{R}^p)$. We will assume that $\Phi$ satisfies the following properties:

- $\Phi$ is **causal**: For any $t \in [0, T]$, if $u, v \in C_0([0, T]; \mathbb{R}^p)$ are two input signals, such that $u|_{[0,t]} \equiv v|_{[0,t]}$, then $\Phi(u)(t) = \Phi(v)(t)$, i.e. the value of $\Phi(u)(t)$ at time $t$ does not depend on future values $\{u(\tau) \,|\, \tau > t\}$.

- $\Phi$ is **continuous** as an operator

$$\Phi : (C_0([0, T]; \mathbb{R}^p), \|\cdot\|_{L^\infty}) \to (C_0([0, T]; \mathbb{R}^q), \|\cdot\|_{L^\infty}),$$

with respect to the $L^\infty$-norm on the input-/output-signals.

Note that the class of Continuous and Causal operators are very general and natural in the contexts of mapping between sequence spaces or time-varying function spaces, see [7, 6] and references therein.

## 3.2 Universal approximation Theorem

The universality of neural oscillators is summarized in the following theorem:

***Theorem* 3.1.** [Universality of the multi-layer neural oscillator] Let $\Phi : C_0([0, T]; \mathbb{R}^p) \to C_0([0, T]; \mathbb{R}^q)$ be a causal and continuous operator. Let $K \subset C_0([0, T]; \mathbb{R}^p)$ be compact. Then for any $\epsilon > 0$, there exist hyperparameters $L, m_1, \ldots, m_L$, weights $w^\ell \in \mathbb{R}^{m_\ell}$, $V^\ell \in \mathbb{R}^{m_\ell \times m_{\ell-1}}$, $A \in \mathbb{R}^{q \times m_L}$ and bias vectors $b^\ell \in \mathbb{R}^{m_\ell}$, $c \in \mathbb{R}^q$, for $\ell = 1, \ldots, L$, such that the output $z : [0, T] \to \mathbb{R}^q$ of the multi-layer neural oscillator (2.2) satisfies

$$\sup_{t \in [0,T]} |\Phi(u)(t) - z(t)| \le \epsilon, \quad \forall u \in K.$$

It is important to observe that the *sparse, independent* multi-layer neural oscillator (2.2) suffices for universality in the considered class. Thus, there is no need to consider the wider class of neural oscillators (2.1), at least in this respect. We remark in passing that Theorem 3.1 immediately implies another universality result for neural oscillators, showing that they can also be used to approximate arbitrary continuous functions $F : \mathbb{R}^p \to \mathbb{R}^q$. This extension is explained in detail in **SM** A.

***Remark* 3.2.** We note that the theorem can be readily extended to remove the requirement on $u(0) = 0$ and $\Phi(u)(0) = 0$. To this end, let $\Phi : C([0, T]; \mathbb{R}^p) \to C([0, T]; \mathbb{R}^q)$ be an operator between spaces of continuous functions, $u \mapsto \Phi(u)$ on $[0, T]$. Fix a $t_0 > 0$, and extend any input function $u : [0, T] \to \mathbb{R}^p$ to a function $\mathcal{E}(u) \in C_0([-t_0, T]; \mathbb{R}^p)$, by

$$\mathcal{E}(u)(t) := \begin{cases} \frac{(t_0+t)}{t_0} u(0), & t \in [-t_0, 0), \\ u(t), & t \in [0, T]. \end{cases}$$

Our proof of Theorem 3.1 can readily be used to show that the oscillator system with forcing $\mathcal{E}(u)$, and initialized at time $-t_0 < 0$, can uniformly approximate $\Phi(u)$ over the entire time interval $[0, T]$, without requiring that $u(0) = 0$, or $\Phi(u)(0) = 0$. In this case, the initial time interval $[-t_0, 0]$ provides the required "warm-up phase" for the neural oscillator.

***Remark* 3.3.** The required compactness property of the set of input signals $K \subset C_0([0, T]; \mathbb{R}^p)$ in Theorem 3.1 is implied by a suitable smoothness constraint; examples include uniform bounds on the Lipschitz norm, Hölder norms, uniform bounds on higher-order derivatives, or under the assumption that the input signals are uniformly bounded and band-limited. The proposed compactness assumption subsumes all of these smoothness constraints, resulting in a widely applicable universal approximation theorem.

***Remark* 3.4.** In practice, neural ODEs such as (2.2) need to be discretized via suitable numerical schemes. As examples, CoRNN and UnICORNN were implemented in [27] and [28], respectively, with implicit-explicit time discretizations. Nevertheless, universality also applies for such discretizations as long as the time-step is small enough, as the underlying discretization is going to be a sufficiently accurate approximation of (2.2) and Theorem 3.1 can be used for showing universality of the discretized version of the multi-layer neural oscillator (2.2).

## 3.3 Outline of the Proof

In the following, we outline the proof of the universality Theorem 3.1, while postponing the technical details to the **SM**. For a given tolerance $\epsilon$, we will explicitly construct the weights and biases of the multi-layer neural oscillator (2.2) such that the underlying operator can be approximated within the given tolerance. This construction takes place in the following steps:

**(Forced) Harmonic Oscillators compute a time-windowed sine transform.** Recall that the forced harmonic oscillator (2.6) is the simplest example of a neural oscillator (2.1). The following lemma, proved by direct calculation in **SM** B.1, shows that this forced harmonic oscillator actually computes a time-windowed variant of the sine transform at the corresponding frequency:

**Lemma 3.5.** Assume that $\omega \neq 0$. Then the solution of (2.6) is given by

$$y(t) = \frac{1}{\omega} \int_0^t u(t-\tau) \sin(\omega\tau)\, d\tau. \qquad (3.1)$$

Given the last result, for a function $u$, we define its **time-windowed sine transform** as follows,

$$\mathcal{L}_t u(\omega) := \int_0^t u(t-\tau) \sin(\omega\tau)\, d\tau. \qquad (3.2)$$

Lemma 3.5 shows that a forced harmonic oscillator computes (3.2) up to a constant.

Figure 1: Illustration of the universal 3-layer neural oscillator architecture constructed in the proof of Theorem 3.1.

**Approximation of causal operators from finite realizations of time-windowed sine transforms.** The following novel result, termed the *fundamental Lemma*, shows that the time-windowed sine transform (3.2) composed with a suitable nonlinear function can approximate causal operators $\Phi$ to desired accuracy; as a consequence, one can conclude that *forced harmonic oscillators* combined with a *nonlinear read-out* defines a universal architecture in the sense of Theorem 3.1.

**Lemma 3.6** (Fundamental Lemma). Let $\Phi : K \subset C_0([0,T];\mathbb{R}^p) \to C_0([0,T];\mathbb{R}^q)$ be a causal and continuous operator, with $K \subset C_0([0,T];\mathbb{R}^p)$ compact. Then for any $\epsilon > 0$, there exists $N \in \mathbb{N}$, frequencies $\omega_1, \ldots, \omega_N$ and a continuous mapping $\Psi : \mathbb{R}^{p \times N} \times [0, T^2/4] \to \mathbb{R}^q$, such that

$$|\Phi(u)(t) - \Psi(\mathcal{L}_t u(\omega_1), \ldots, \mathcal{L}_t u(\omega_N); t^2/4)| \leq \epsilon,$$

for all $u \in K$ and $t \in [0,T]$.

The function $\Psi$ defined in Lemma 3.6 can be interpreted as a finite-dimensional encoding of the operator $\Phi$. The main insight of the fundamental Lemma 3.6 is that the (infinite-dimensional) operator $\Phi$ can effectively be replaced by a (finite-dimensional) function $\Psi$, making the approximation by neural oscillators more tractable.

The proof of the fundamental Lemma 3.6, detailed in **SM** B.2, is based on first showing that any continuous function can be reconstructed to desired accuracy, in terms of realizations of its time-windowed sine transform (3.2) at finitely many frequencies $\omega_1, \ldots, \omega_N$ (see **SM** Lemma B.1). Then, we leverage the continuity of the underlying operator $\Phi$ to approximate it with a finite-dimensional function $\Psi$, which takes the time-windowed sine transforms as its arguments.

Given these two results, we can discern a clear strategy to prove the universality Theorem 3.1. First, we will show that a general *nonlinear* form of the neural oscillator (2.2) can also compute the time-windowed sine transform at arbitrary frequencies. Then, these outputs need to be processed in order to apply the fundamental Lemma 3.6 and approximate the underlying operator $\Phi$. To this end, we will also approximate the function $\Psi$ (mapping finite-dimensional inputs to finite-dimensional outputs) by oscillatory layers. The concrete steps in this strategy are outlined below.

**Nonlinear Oscillators approximate the time-windowed sine transform.** The building block of multi-layer neural oscillators (2.2) is the nonlinear oscillator of the form,

$$\ddot{y} = \sigma(w \odot y + Vu + b). \tag{3.3}$$

In the following Lemma (proved in **SM** B.3), we show that even for a nonlinear activation function $\sigma$ such as $\tanh$ or $\sin$, the nonlinear oscillator (3.3) can approximate the time-windowed sine transform.

***Lemma 3.7.*** Fix $\omega \neq 0$. Assume that $\sigma(0) = 0$, $\sigma'(0) = 1$. For any $\epsilon > 0$, there exist $w, V, b, A \in \mathbb{R}$, such that the nonlinear oscillator (3.3), initialized at $y(0) = \dot{y}(0) = 0$, has output

$$|Ay(t) - \mathcal{L}_t u(\omega)| \leq \epsilon, \qquad \forall u \in K, \ t \in [0, T],$$

with $\mathcal{L}_t u(\omega)$ being the time-windowed sine transform (3.2).

**Coupled Nonlinear Oscillators approximate time-delays.** The next step in the proof is to show that coupled oscillators can approximate time-delays in the continuous input signal. This fact will be of crucial importance in subsequent arguments. We have the following Lemma (proved in **SM** B.4),

***Lemma 3.8.*** Let $K \subset C_0([0, T]; \mathbb{R}^p)$ be a compact subset. For every $\epsilon > 0$, and $\Delta t \geq 0$, there exist $m \in \mathbb{N}$, $w \in \mathbb{R}^m$, $V \in \mathbb{R}^{m \times p}$, $b \in \mathbb{R}^m$ and $A \in \mathbb{R}^{p \times m}$, such that the oscillator (3.3), initialized at $y(0) = \dot{y}(0) = 0$, has output

$$\sup_{t \in [0,T]} |u(t - \Delta t) - Ay(t)| \leq \epsilon, \quad \forall u \in K,$$

where $u(t)$ is extended to negative values $t < 0$ by zero.

**Two-layer neural oscillators approximate neural networks pointwise.** As in the strategy outlined above, the final ingredient in our proof of the universality theorem 3.1 is to show that neural oscillators can approximate continuous functions, such as the $\Psi$ in the fundamental lemma 3.6, to desired accuracy. To this end, we will first show that neural oscillators can approximate general neural networks (perceptrons) and then use the universality of neural networks in the class of continuous functions to prove the desired result. The main difficulty here is that, due to the underlying oscillatory dynamics, it is not clear whether the output of even a shallow neural network can be reproduced by neural oscillators. The following lemma, visually represented by the bottom half of Figure 1, guarantees this:

***Lemma 3.9.*** Let $K \subset C_0([0, T]; \mathbb{R}^p)$ be compact. For matrices $\Sigma, \Lambda$ and bias $\gamma$, and any $\epsilon > 0$, there exists a two-layer ($L = 2$) oscillator (2.2), initialized at $y^\ell(0) = \dot{y}^\ell(0) = 0$, $\ell = 1, 2$, such that

$$\sup_{t \in [0,T]} \left| \left[ Ay^2(t) + c \right] - \Sigma\sigma(\Lambda u(t) + \gamma) \right| \leq \epsilon, \quad \forall u \in K.$$

The proof, detailed in **SM** B.5, is constructive and the neural oscillator that we construct has two layers. The first layer just processes a nonlinear input function through a nonlinear oscillator and the second layer, approximates the second-derivative (in time) from time-delayed versions of the input signal that were constructed in Lemma 3.8.

**Combining the ingredients to prove the universality theorem 3.1.** The afore-constructed ingredients are combined in **SM** B.6 to prove the universality theorem. In this proof, we explicitly construct a *three-layer* neural oscillator (2.2) which approximates the underlying operator $\Phi$. The first layer follows the construction of Lemma 3.7, to approximate the time-windowed sine transform (3.2), for as many frequencies as are required in the fundamental Lemma 3.6. The second- and third-layers imitate the construction of Lemma 3.9 to approximate a neural network (perception), which in turn by the universal approximation of neural networks, approximates the function $\Psi$ in Lemma 3.6 to desired accuracy. Putting the network together leads to a three-layer oscillator that approximates the continuous and casual operator $\Phi$. This construction is depicted in Figure 1.

## 4 Discussion

Machine learning architectures, based on networks of coupled oscillators, for instance sequence models such as CoRNN [27] and UnICORNN [28], graph neural networks such as GraphCON [26] and increasingly, the so-called physical neural networks (PNNs) such as linear and nonlinear mechanical oscillators [35] and spintronic oscillators [24, 32], are being increasingly used. A priori, it is unclear why ML systems based on oscillators can provide competitive performance on a variety of learning benchmarks, e.g. [27, 28, 26, 35], rather than biasing their outputs towards oscillatory functions. In order to address these concerns about their expressivity, we have investigated the theoretical properties of machine learning systems based on oscillators. Our main aim was to answer a fundamental question: *"are coupled oscillator based machine learning architectures universal?"*. In other words, can these architectures, in principle, approximate a large class of input-output maps to desired accuracy.

To answer this fundamental question, we introduced an abstract framework of *neural oscillators* (2.1) and its particular instantiation, the *multi-layer neural oscillators* (2.2). This abstract class of *second-order neural ODEs* encompasses both sequence models such as CoRNN and UnICORNN, as well as a very general and representative PNN, based on the so-called Frenkel-Kontorova model. The main contribution of this paper was to prove the universality theorem 3.1 on the ability of multi-layer neural oscillators (2.2) to approximate a large class of operators, namely causal and continuous maps between spaces of continuous functions, to desired accuracy. Despite the fact that the considered neural oscillators possess a very specific and constrained structure, not even encompassing general Hamiltonian systems, the approximated class of operators is nevertheless very general, including solution operators of general ordinary and even time-delay differential equations.

The crucial theoretical ingredient in our proof was the *fundamental Lemma* 3.6, which implies that linear oscillator dynamics combined with a pointwise nonlinear read-out suffices for universal operator approximation; our construction can correspondingly be thought of as a large number of linear processors, coupled with nonlinear readouts. This construction could have implications for other models such as *structured state space models* [9, 8, 10] which follow a similar paradigm, and the extension of our universality results to such models could be of great interest.

Our universality result has many interesting implications. To start with, we rigorously prove that an ML architecture based on coupled oscillators can approximate a very large class of operators. This provides theoretical support to many widely used sequence models and PNNs based on oscillators. Moreover, given the generality of our result, we hope that such a universality result can spur the design of innovative architectures based on oscillators, particularly in the realm of analog devices as ML inference systems or ML accelerators [35].

It is also instructive to lay out some of the limitations of the current article and point to avenues for future work. In this context, our setup currently only considers time-varying functions as inputs and outputs. Roughly speaking, these inputs and outputs have the structure of (infinite) sequences. However, a large class of learning tasks can be reconfigured to take sequential inputs and outputs. These include text (as evident from the tremendous success of large language models [22]), DNA sequences, images [16], timeseries and (offline) reinforcement learning [15]. Nevertheless, a next step would be to extend such universality results to inputs (and outputs) which have some spatial or relational structure, for instance by considering functions which have a spatial dependence or which are defined on graphs. On the other hand, the class of operators that we consider, i.e., casual and continuous, is not only natural in this setting but very general [7, 6].

Another limitation lies in the *feed forward* structure of the multi-layer neural oscillator (2.2). As mentioned before, most physical (and neurobiological) systems exhibit feedback loops between their constituents. However, this is not common in ML systems. In fact, we had to use a *mass ordering* in the Frenkel-Kontorova system of coupled pendula (2.8) in order to recast it in the form of the multi-layer neural oscillator (2.2). Such asymptotic ordering may not be possible for arbitrary physical neural networks. Exploring how such ordering mechanisms might arise in physical and biological systems in order to effectively give rise to a feed forward system could be very interesting. One possible mechanism for coupled oscillators that can lead to a hierarchical structure is that of synchronization [34, 31] and references therein. How such synchronization interacts with universality is a very interesting question and will serve as an avenue for future work.

Finally, universality is arguably necessary but far from sufficient to analyze the performance of any ML architecture. Other aspects such as trainability and generalization are equally important, and we do not address these issues here. We do mention that trainability of oscillatory systems would profit from the fact that oscillatory dynamics is (gradient) stable and this formed the basis of the proofs of mitigation of the exploding and vanishing gradient problem for CoRNN in [27] and UnICORNN in [28] as well as GraphCON in [26]. Extending these results to the general second-order neural ODE (2.2), for instance through an analysis of the associated adjoint system, is left for future work.

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

# A  Another universality result for neural oscillators

The universal approximation Theorem 3.1 immediately implies another universal approximation results for neural oscillators, as explained next. We consider a continuous map $F : \mathbb{R}^p \to \mathbb{R}^q$; our goal is to show that $F$ can be approximated to given accuracy $\epsilon$ by suitably defined neural oscillators. Fix a time interval $[0, T]$ for (an arbitrary choice) $T = 2$. Let $K_0 \subset \mathbb{R}^p$ be a compact set. Given $\xi \in \mathbb{R}^p$, we associate with it a function $u_\xi(t) \in C_0([0, T]; \mathbb{R}^p)$, by setting

$$u_\xi(t) := t\xi. \tag{A.1}$$

Clearly, the set $K := \{u_\xi \,|\, \xi \in K_0\}$ is compact in $C_0([0, T]; \mathbb{R}^p)$. Furthermore, we can define an operator $\Phi : C_0([0, T]; \mathbb{R}^p) \to C_0([0, T]; \mathbb{R}^q)$, by

$$\Phi(u)(t) := \begin{cases} 0, & t \in [0, 1), \\ (t - 1)F(u(1)), & t \in [1, T]. \end{cases} \tag{A.2}$$

where $F : \mathbb{R}^p \to \mathbb{R}^q$ is the given continuous function that we wish to approximate. One readily checks that $\Phi$ defines a causal and continuous operator. Note, in particular, that

$$\Phi(u_\xi)(T) = (T - 1)F(u_\xi(1)) = F(\xi),$$

is just the evaluation of $F$ at $\xi$, for any $\xi \in K_0$.

Since neural oscillators can uniformly approximate the operator $\Phi$ for inputs $u_\xi \in K$, then as a consequence of Theorem 3.1 and (2.3), it follows that, for any $\epsilon > 0$ there exists $m \in \mathbb{N}$, matrices $W \in \mathbb{R}^{m \times m}$, $V \in \mathbb{R}^{m \times p}$ and $A \in \mathbb{R}^{q \times m}$, and bias vectors $b \in \mathbb{R}^m$, $c \in \mathbb{R}^q$, such that for any $\xi \in K_0$, the neural oscillator system,

$$\begin{cases} \ddot{y}_\xi(t) = \sigma\left(W y_\xi(t) + tV\xi + b\right), & \text{(A.3)} \\ y_\xi(0) = \dot{y}_\xi(0) = 0, & \text{(A.4)} \\ z_\xi(t) = A y_\xi(t) + c, & \text{(A.5)} \end{cases}$$

satisfies

$$|z_\xi(T) - F(\xi)| = |z_\xi(T) - \Phi(u_\xi)(T)| \leq \sup_{t \in [0,T]} |z_\xi(t) - \Phi(u_\xi)(t)| \leq \epsilon,$$

uniformly for all $\xi \in K_0$. Hence, neural oscillators can be used to approximate an arbitrary continuous function $F : \mathbb{R}^p \to \mathbb{R}^q$, uniformly over compact sets. Thus, neural oscillators also provide universal function approximation.

# B  Proof of Theorem 3.1

## B.1  Proof of Lemma 3.5

*Proof.* We can rewrite $y(t) = \frac{1}{\omega} \int_0^t u(\tau) \sin(\omega(t - \tau)) \, d\tau$. By direct differentiation, one readily verifies that $y(t)$ so defined, satisfies

$$\dot{y}(t) = \int_0^t u(\tau) \cos(\omega(t - \tau)) \, d\tau + [u(\tau) \sin(\omega(t - \tau))]_{\tau=t} = \int_0^t u(\tau) \cos(\omega(t - \tau)) \, d\tau,$$

in account of the fact that $\sin(0) = 0$. Differentiating once more, we find that

$$\ddot{y}(t) = -\omega \int_0^t u(\tau) \sin(\omega(t - \tau)) \, d\tau + [u(\tau) \cos(\omega(t - \tau))]_{\tau=t}$$
$$= -\omega^2 y(t) + u(t).$$

Thus $y(t)$ solves the ODE (2.6), with initial condition $y(0) = \dot{y}(0) = 0$. □

## B.2 Proof of Fundamental Lemma 3.6

**Reconstruction of a continuous signal from its sine transform.** Let $[0, T] \subset \mathbb{R}$ be an interval. We recall that we define the windowed sine transform $\mathcal{L}_t u(\omega)$ of a function $u : [0, T] \to \mathbb{R}^p$, by

$$\mathcal{L}_t u(\omega) = \int_0^t u(t - \tau) \sin(\omega \tau)\, d\tau, \quad \omega \in \mathbb{R}.$$

In the following, we fix a compact set $K \subset C_0([0, T]; \mathbb{R}^p)$. Note that for any $u \in K$, we have $u(0) = 0$, and hence $K$ can be identified with a subset of $C((-\infty, T]; \mathbb{R}^p)$, consisting of functions with $\operatorname{supp}(u) \subset [0, T]$. We consider the reconstruction of continuous functions $u \in K$. We will show that $u$ can be approximately reconstructed from knowledge of $\mathcal{L}_t(\omega)$. More precisely, we provide a detailed proof of the following result:

***Lemma* B.1.** Let $K \subset C((-\infty, T]; \mathbb{R}^p)$ be compact, such that $\operatorname{supp}(u) \subset [0, T]$ for all $u \in K$. For any $\epsilon, \Delta t > 0$, there exists $N \in \mathbb{N}$, frequencies $\omega_1, \ldots, \omega_N \in \mathbb{R} \setminus \{0\}$, phase-shifts $\vartheta_1, \ldots, \vartheta_N \in \mathbb{R}$ and weights $\alpha_1, \ldots, \alpha_N \in \mathbb{R}$, such that

$$\sup_{\tau \in [0, \Delta t]} \left| u(t - \tau) - \sum_{j=1}^N \alpha_j \mathcal{L}_t u(\omega_j) \sin(\omega_j \tau - \vartheta_j) \right| \le \epsilon,$$

for all $u \in K$ and for all $t \in [0, T]$.

*Proof.* **Step 0: (Equicontinuity)** We recall the following fact from topology. If $K \subset C((-\infty, T]; \mathbb{R}^p)$ is compact, then it is equicontinuous; i.e. there exists a continuous modulus of continuity $\phi : [0, \infty) \to [0, \infty)$ with $\phi(r) \to 0$ as $r \to 0$, such that

$$|u(t - \tau) - u(t)| \le \phi(\tau), \quad \forall \tau \ge 0,\ t \in [0, T],\ \forall u \in K. \tag{B.1}$$

**Step 1: (Connection to Fourier transform)** Fix $t_0 \in [0, T]$ and $u \in K$ for the moment. Define $f(\tau) = u(t_0 - \tau)$. Note that $f \in C([0, \infty); \mathbb{R}^p)$, and $f$ has compact support $\operatorname{supp}(f) \subset [0, T]$. We also note that, by (B.1), we have

$$|f(t + \tau) - f(t)| \le \phi(\tau), \quad \forall \tau \ge 0,\ t \in [0, T].$$

We now consider the following odd extension of $f$ to all of $\mathbb{R}$:

$$F(\tau) := \begin{cases} f(\tau), & \text{for } \tau \ge 0, \\ -f(-\tau), & \text{for } \tau \le 0. \end{cases}$$

Since $F$ is odd, the Fourier transform of $F$ is given by

$$\widehat{F}(\omega) := \int_{-\infty}^\infty F(\tau) e^{-i\omega\tau}\, d\tau = i \int_{-\infty}^\infty F(\tau) \sin(\omega\tau)\, d\tau = 2i \int_0^T f(\tau) \sin(\omega\tau)\, d\tau = 2i\mathcal{L}_{t_0} u(\omega).$$

Let $\epsilon > 0$ be arbitrary. Our goal is to uniformly approximate $F(\tau)$ on the interval $[0, \Delta t]$. The main complication here is that $F$ lacks regularity (is discontinuous), and hence the inverse Fourier transform of $\widehat{F}$ does not converge to $F$ uniformly over this interval; instead, a more careful reconstruction based on mollification of $F$ is needed. We provide the details below.

**Step 2: (Mollification)** We now fix a smooth, non-negative and compactly supported function $\rho : \mathbb{R} \to \mathbb{R}$, such that $\operatorname{supp}(\rho) \subset [0, 1]$, $\rho \ge 0$, $\int_{\mathbb{R}} \rho(t)\, dt = 1$, and we define a mollifier $\rho_\epsilon(t) := \frac{1}{\epsilon}\rho(t/\epsilon)$. In the following, we will assume throughout that $\epsilon \le T$. We point out that $\operatorname{supp}(\rho_\epsilon) \subset [0, \epsilon]$, and hence, the mollification $F_\epsilon(t) = (F * \rho_\epsilon)(t)$ satisfies, for $t \ge 0$:

$$|F(t) - F_\epsilon(t)| = \left| \int_0^\epsilon (F(t) - F(t + \tau))\rho_\epsilon(\tau)\, d\tau \right| = \left| \int_0^\epsilon (f(t) - f(t + \tau))\rho_\epsilon(\tau)\, d\tau \right|$$

$$\le \left\{ \sup_{\tau \in [0, \epsilon]} |f(t) - f(t + \tau)| \right\} \int_0^\epsilon \rho_\epsilon(\tau)\, d\tau \le \phi(\epsilon).$$

In particular, this shows that

$$\sup_{t \in [0,T]} |F(t) - F_\epsilon(t)| \le \phi(\epsilon),$$

can be made arbitrarily small, with an error that depends only on the modulus of continuity $\phi$.

**Step 3: (Fourier inverse)** Let $\widehat{F}_\epsilon(\omega)$ denote the Fourier transform of $F_\epsilon$. Since $F_\epsilon$ is smooth and compactly supported, it is well-known that we have the identity

$$F_\epsilon(\tau) = \frac{1}{2\pi} \int_{-\infty}^{\infty} \widehat{F}_\epsilon(\omega) e^{-i\omega\tau} \, d\omega, \qquad \forall\, t \in \mathbb{R},$$

where $\omega \mapsto \widehat{F}_\epsilon(\omega)$ decays to zero very quickly (almost exponentially) as $|\omega| \to \infty$. In fact, since $F_\epsilon = F * \rho_\epsilon$ is a convolution, we have $\widehat{F}_\epsilon(\omega) = \widehat{F}(\omega)\widehat{\rho}_\epsilon(\omega)$, where $|\widehat{F}(\omega)| \le 2\|f\|_{L^\infty} T$ is uniformly bounded, and $\widehat{\rho}_\epsilon(\omega)$ decays quickly. In particular, this implies that there exists a $L = L(\epsilon, T) > 0$ *independent of $f$*, such that

$$\left| F_\epsilon(\tau) - \frac{1}{2\pi} \int_{-L}^{L} \widehat{F}(\omega)\widehat{\rho}_\epsilon(\omega) e^{-i\omega\tau} \, d\omega \right| \le 2T\|f\|_{L^\infty} \int_{|\omega|>L} |\widehat{\rho}_\epsilon(\omega)| \, d\omega \le \|f\|_{L^\infty}\epsilon, \qquad \forall\, \tau \in \mathbb{R}.$$

(B.2)

**Step 4: (Quadrature)** Next, we observe that, since $F$ and $\rho_\epsilon$ are compactly supported, their Fourier transform $\omega \mapsto \widehat{F}(\omega)\widehat{\rho}_\epsilon(\omega) e^{-i\omega\tau}$ is smooth; in fact, for $|\tau| \le T$, the Lipschitz constant of this mapping can be explicitly estimated by noting that

$$\frac{\partial}{\partial\omega}\left[ \widehat{F}(\omega)\widehat{\rho}_\epsilon(\omega) e^{-i\omega\tau} \right] = \frac{\partial}{\partial\omega} \int_{\mathrm{supp}(F_\epsilon)} (F * \rho_\epsilon)(t) e^{i\omega(t-\tau)} \, dt$$

$$= \int_{\mathrm{supp}(F_\epsilon)} i(t-\tau)(F * \rho_\epsilon)(t) e^{i\omega(t-\tau)} \, dt.$$

We next take absolute values, and note that any $t$ in the support of $F_\epsilon$ obeys the bound $|t| \le T + \epsilon \le 2T$, while $|\tau| \le T$ by assumption; it follows that

$$\mathrm{Lip}\left( \omega \mapsto \widehat{F}(\omega)\widehat{\rho}_\epsilon(\omega) e^{-i\omega\tau} \right) \le (2T + T)\|F\|_{L^\infty}\|\rho_\epsilon\|_{L^1} = 3T\|F\|_{L^\infty}, \quad \forall\, \tau \in [0,T].$$

It thus follows from basic results on quadrature that for an equidistant choice of frequencies $\omega_1 < \cdots < \omega_N$, with spacing $\Delta\omega = 2L/(N-1)$, we have

$$\left| \frac{1}{2\pi} \int_{-L}^{L} \widehat{F}(\omega)\widehat{\rho}_\epsilon(\omega) e^{-i\omega\tau} \, d\omega - \frac{\Delta\omega}{2\pi} \sum_{j=1}^{N} \widehat{F}(\omega_j)\widehat{\rho}_\epsilon(\omega_j) e^{-i\omega_j\tau} \right| \le \frac{CL^2\, 3T\|F\|_{L^\infty}}{N}, \quad \forall\, \tau \in [0,T],$$

for an absolute constant $C > 0$, independent of $F$, $T$ and $N$. By choosing $N$ to be even, we can ensure that $\omega_j \ne 0$ for all $j$. In particular, recalling that $L = L(T,\epsilon)$ depends only on $\epsilon$ and $T$, and choosing $N = N(T,\epsilon)$ sufficiently large, we can combine the above estimate with (B.2) to ensure that

$$\left| F_\epsilon(\tau) - \frac{\Delta\omega}{2\pi} \sum_{j=1}^{N} \widehat{F}(\omega_j)\widehat{\rho}_\epsilon(\omega_j) e^{-i\omega_j\tau} \right| \le 2\|f\|_{L^\infty}\epsilon, \quad \forall\, \tau \in [0,T],$$

where we have taken into account that $\|F\|_{L^\infty} = \|f\|_{L^\infty}$.

**Step 5: (Conclusion)** To conclude the proof, we recall that $\widehat{F}(\omega) = 2i\mathcal{L}_{t_0}u(\omega)$ can be expressed in terms of the sine transform $\mathcal{L}_t u$ of the function $u$ which was fixed at the beginning of Step 1. Recall also that $f(\tau) = u(t_0 - \tau)$, so that $\|f\|_{L^\infty} = \|u\|_{L^\infty}$. Hence, we can write the real part of $\frac{\Delta\omega}{2\pi}\widehat{F}(\omega_j)\widehat{\rho}_\epsilon(\omega_j) e^{-i\omega_j\tau} = \frac{\Delta\omega}{2\pi} 2i\mathcal{L}_{t_0}u(\omega_j)\widehat{\rho}_\epsilon(\omega_j) e^{-i\omega_j\tau}$, in the form $\alpha_j \mathcal{L}_{t_0}(\omega_j)\sin(\omega_j\tau - \vartheta_j)$ for coefficients $\alpha_j \in \mathbb{R}$ and $\theta_j \in \mathbb{R}$ which depend only on $\Delta\omega$ and $\widehat{\rho}_\epsilon(\omega_j)$, but are independent of $u$. In

particular, it follows that

$$
\sup_{\tau \in [0,\Delta t]} \left| u(t_0 - \tau) - \sum_{j=1}^{N} \alpha_j \mathcal{L}_{t_0} u(\omega_j) \sin(\omega_j \tau - \vartheta_j) \right| = \sup_{t \in [0,\Delta t]} \left| F(\tau) - \mathrm{Re}\left( \frac{\Delta \omega}{2\pi} \sum_{j=1}^{N} \widehat{F}(\omega_j) \widehat{\rho}_\epsilon(\omega_j) e^{-i\omega_j \tau} \right) \right|
$$

$$
\leq \sup_{\tau \in [0,\Delta t]} \left| F(\tau) - \frac{\Delta \omega}{2\pi} \sum_{j=1}^{N} \widehat{F}(\omega_j) \widehat{\rho}_\epsilon(\omega_j) e^{-i\omega_j \tau} \right|
$$

$$
\leq \sup_{\tau \in [0,\Delta t]} |F(\tau) - F_\epsilon(\tau)|
$$

$$
+ \sup_{\tau \in [0,\Delta t]} \left| F_\epsilon(\tau) - \frac{\Delta \omega}{2\pi} \sum_{j=1}^{N} \widehat{F}(\omega_j) \widehat{\rho}_\epsilon(\omega_j) e^{-i\omega_j \tau} \right|.
$$

By Steps 1 and 3, the first term on the right-hand side is bounded by $\leq \phi(\epsilon)$, while the second one is bounded by $\leq 2 \sup_{u \in K} \|u\|_{L^\infty} \epsilon \leq C\epsilon$, where $C = C(K) < \infty$ depends only on the compact set $K \subset C([0,T]; \mathbb{R}^p)$. Hence, we have

$$
\sup_{\tau \in [0,\Delta t]} \left| u(t_0 - \tau) - \sum_{j=1}^{N} \alpha_j \mathcal{L}_{t_0} u(\omega_j) \sin(\omega_j \tau - \vartheta_j) \right| \leq \phi(\epsilon) + C\epsilon.
$$

In this estimate, the function $u \in K$ and $t_0 \in [0,T]$ were arbitrary, and the modulus of continuity $\phi$ as well as the constant $C$ on the right-hand side depend only on the set $K$. it thus follows that for this choice of $\alpha_j$, $\omega_j$ and $\vartheta_j$, we have

$$
\sup_{u \in K} \sup_{t \in [0,T]} \sup_{\tau \in [0,\Delta t]} \left| u(t - \tau) - \sum_{j=1}^{N} \alpha_j \mathcal{L}_t u(\omega_j) \sin(\omega_j \tau - \vartheta_j) \right| \leq \phi(\epsilon) + C\epsilon.
$$

Since $\epsilon > 0$ was arbitrary, the right-hand side can be made arbitrarily small. The claim then readily follows.

$\square$

The next step in the proof of the fundamental Lemma 3.6 needs the following preliminary result in functional analysis,

**Lemma B.2.** Let $\mathcal{X}, \mathcal{Y}$ be Banach spaces, and let $K \subset \mathcal{X}$ be a compact subset. Assume that $\Phi : \mathcal{X} \to \mathcal{Y}$ is continous. Then for any $\epsilon > 0$, there exists a $\delta > 0$, such that if $\|u - u^K\|_{\mathcal{X}} \leq \delta$ with $u \in \mathcal{X}$, $u^K \in K$, then $\|\Phi(u) - \Phi(u^K)\|_{\mathcal{Y}} \leq \epsilon$.

*Proof.* Suppose not. Then there exists $\epsilon_0 > 0$ and a sequence $u_j, u_j^K$, ($j \in \mathbb{N}$), such that $\|u_j - u_j^K\|_{\mathcal{X}} \leq j^{-1}$, while $\|\Phi(u_j) - \Phi(u_j^K)\|_{\mathcal{Y}} \geq \epsilon_0$. By the compactness of $K$, we can extract a subsequence $j_k \to \infty$, such that $u_{j_k}^K \to u^K$ converges to some $u^K \in K$. By assumption on $u_j$, this implies that

$$
\|u_{j_k} - u^K\|_{\mathcal{X}} \leq \|u_{j_k} - u_{j_k}^K\|_{\mathcal{X}} + \|u_{j_k}^K - u^K\|_{\mathcal{X}} \overset{(k \to \infty)}{\longrightarrow} 0,
$$

which, by the assumed continuity of $\Phi$, leads to the contradiction that $0 < \epsilon_0 \leq \|\Phi(u_{j_k}) - \Phi(u^K)\|_{\mathcal{Y}} \to 0$, as $k \to \infty$. $\square$

**Proof of Lemma 3.6.** Now, we can prove the fundamental Lemma in the following,

*Proof.* Let $\epsilon > 0$ be given. We can identify $K \subset C_0([0,T]; \mathbb{R}^p)$ with a compact subset of $C((-\infty, T]; \mathbb{R}^p)$, by extending all $u \in K$ by zero for negative times, i.e. we set $u(t) = 0$ for $t < 0$. Applying Lemma B.2, with $\mathcal{X} = C_0([0,T]; \mathbb{R}^p)$ and $\mathcal{Y} = C_0([0,T]; \mathbb{R}^q)$, we can find a $\delta > 0$, such that for any $u \in C_0([0,T]; \mathbb{R}^p)$ and $u^K \in K$, we have

$$
\|u - u^K\|_{L^\infty} \leq \delta \quad \Rightarrow \quad \|\Phi(u) - \Phi(u^K)\|_{L^\infty} \leq \epsilon. \tag{B.3}
$$

By the inverse sine transform Lemma B.1, there exist $N \in \mathbb{N}$, frequencies $\omega_1, \ldots, \omega_N \neq 0$, phase-shifts $\vartheta_1, \ldots, \vartheta_N$ and coefficients $\alpha_1, \ldots, \alpha_N$, such that for any $u \in K$ and $t \in [0, T]$:

$$\sup_{\tau \in [0,T]} \left| u(t - \tau) - \sum_{j=1}^{N} \alpha_j \, \mathcal{L}_t u(\omega_j) \sin(\omega_j \tau - \vartheta_j) \right| \leq \delta.$$

Given $\mathcal{L}_t u(\omega_1), \ldots, \mathcal{L}_t u(\omega_N)$, we can thus define a reconstruction mapping $\mathcal{R} : \mathbb{R}^N \times [0, T] \to C([0, T]; \mathbb{R}^p)$ by

$$\mathcal{R}(\beta_1, \ldots, \beta_N; t)(\tau) := \sum_{j=1}^{N} \alpha_j \beta_j \sin(\omega_j (t - \tau) - \vartheta_j).$$

Then, for $\tau \in [0, t]$, we have

$$|u(\tau) - \mathcal{R}(\mathcal{L}_t u(\omega_1), \ldots, \mathcal{L}_t u(\omega_N); t)(\tau)| \leq \delta.$$

We can now uniquely define $\Psi : \mathbb{R}^N \times [0, T^2/4] \to C_0([0, T]; \mathbb{R}^p)$, by the identity

$$\Psi(\mathcal{L}_t u(\omega_1), \ldots, \mathcal{L}_t u(\omega_N); t^2/4) = \Phi\left(\mathcal{R}(\mathcal{L}_t u(\omega_1), \ldots, \mathcal{L}_t u(\omega_N); t)\right).$$

Using the short-hand notation $\mathcal{R}_t u = \mathcal{R}(\mathcal{L}_t u(\omega_1), \ldots, \mathcal{L}_t u(\omega_N); t)$, we have $\sup_{\tau \in [0,t]} |u(\tau) - \mathcal{R}_t u(\tau)| \leq \delta$, for all $t \in [0, T]$. By (B.3), this implies that

$$\left| \Phi(u)(t) - \Psi(\mathcal{L}_t u(\omega_1), \ldots, \mathcal{L}_t u(\omega_N); t^2/4) \right| = |\Phi(u)(t) - \Phi(\mathcal{R}_t u)(t)| \leq \epsilon.$$

$\square$

### B.3 Proof of Lemma 3.7

*Proof.* Let $\omega \neq 0$ be given. For a (small) parameter $s > 0$, we consider

$$\ddot{y}_s = \frac{1}{s} \sigma(-s\omega^2 y_s + su), \quad y_s(0) = \dot{y}_s(0) = 0.$$

Let $Y$ be the solution of

$$\ddot{Y} = -\omega^2 Y + u, \quad Y(0) = \dot{Y}(0) = 0.$$

Then we have, on account of $\sigma(0) = 0$ and $\sigma'(0) = 1$,

$$
\begin{aligned}
s^{-1}\sigma(-s\omega^2 Y + su) - [-\omega^2 Y + u] &= \frac{\sigma(-s\omega^2 Y + su) - \sigma(0)}{s} - \sigma'(0)[-\omega^2 Y + u] \\
&= \frac{1}{s} \int_0^s \frac{\partial}{\partial \zeta} \left[ \sigma(-\zeta \omega^2 Y + \zeta u) \right] d\zeta - \sigma'(0)[-\omega^2 Y + u] \\
&= \frac{1}{s} \left( \int_0^s \left[ \sigma'(-\zeta \omega^2 Y + \zeta u) - \sigma'(0) \right] d\zeta \right) \left[ -\omega^2 Y + u \right].
\end{aligned}
$$

It follows from Lemma 3.5 that for any input $u \in K$, with $\sup_{u \in K} \|u\|_{L^\infty} =: B < \infty$, we have a uniform bound $\|Y\|_{L^\infty} \leq BT/\omega$, hence we can estimate

$$| - \omega^2 Y + u| \leq B(\omega T + 1),$$

uniformly for all such $u$. In particular, it follows that

$$\left| s^{-1}\sigma(-s\omega^2 Y + su) - [-\omega^2 Y + u] \right| \leq B(T\omega + 1) \sup_{|x| \leq sB(T\omega+1)} |\sigma'(x) - \sigma'(0)|.$$

Clearly, for any $\delta > 0$, we can choose $s \in (0, 1]$ sufficiently small, such that the right hand-side is bounded by $\delta$, i.e. with this choice of $s$,

$$\left| s^{-1}\sigma(-s\omega^2 Y(t) + su(t)) - [-\omega^2 Y(t) + u(t)] \right| \leq \delta, \quad \forall t \in [0, T],$$

holds for any choice of $u \in K$. We will fix this choice of $s$ in the following, and write $g(y, u) := s^{-1}\sigma(-s\omega^2 y + su)$. We note that $g$ is Lipschitz continuous in $y$, for all $|y| \leq BT/\omega$ and $|u| \leq B$, with $\text{Lip}_y(g) \leq \omega^2 \sup_{|\xi| \leq B(\omega T + 1)} |\sigma'(\xi)|$.

To summarize, we have shown that $Y$ solves

$$\ddot{Y} = g(Y, u) + f, \qquad Y(0) = \dot{Y}(0) = 0,$$

where $\|f\|_{L^\infty} \le \delta$. By definition, $y_s$ solves

$$\ddot{y}_s = g(y_s, u), \qquad y_s(0) = \dot{y}_s(0) = 0.$$

It follows from this that

$$|y_s(t) - Y(t)| \le \int_0^t \int_0^\tau \left\{ |g(y_s(\theta), u(\theta)) - g(Y(\theta), u(\theta))| + |f(\theta)| \right\} d\theta \, d\tau$$

$$\le \int_0^t \int_0^\tau \left\{ \mathrm{Lip}_y(g) |y_s(\theta) - Y(\theta)| + \delta \right\} d\theta \, d\tau$$

$$\le T\omega^2 \sup_{|\xi| \le B(\omega T + 1)} |\sigma'(\xi)| \int_0^t |y_s(\tau) - Y(\tau)| \, d\tau + T^2 \delta.$$

Recalling that $Y(t) = \mathcal{L}_t u(\omega)$, then by Gronwall's inequality, the last estimate implies that

$$\sup_{t \in [0,T]} |y_s(t) - \mathcal{L}_t u(\omega)| = \sup_{t \in [0,T]} |y_s - Y| \le C\delta,$$

for a constant $C = C(T, \omega, \sup_{|\xi| \le B(\omega T + 1)} |\sigma'(\xi)|) > 0$, depending only on $T$, $\omega$, $B$ and $\sigma'$. Since $\delta > 0$ was arbitrary, we can ensure that $C\delta \le \epsilon$. Thus, we have shown that a suitably rescaled nonlinear oscillator approximates the harmonic oscillator to any desired degree of accuracy, and uniformly for all $u \in K$.

To finish the proof, we observe that $y$ solves

$$\ddot{y} = \sigma(-\omega^2 y + su), \qquad y(0) = \dot{y}(0) = 0,$$

if, and only if, $y_s = y/s$ solves

$$\ddot{y}_s = s^{-1} \sigma(-s\omega^2 y_s + su), \qquad y_s(0) = \dot{y}_s(0) = 0.$$

Hence, with $W = -\omega^2$, $V = s$, $b = 0$ and $A = s^{-1}$, we have

$$\sup_{t \in [0,T]} |Ay(t) - \mathcal{L}_t u(\omega)| = \sup_{t \in [0,T]} |y_s(t) - \mathcal{L}_t u(\omega)| \le \epsilon.$$

This concludes the proof. $\qquad\qquad\qquad\qquad\qquad\qquad\qquad\qquad\qquad\qquad\qquad\square$

## B.4 Proof of Lemma 3.8

*Proof.* Let $\epsilon, \Delta t$ be given. By the sine transform reconstruction Lemma B.1, there exists $N \in \mathbb{N}$, frequencies $\omega_1, \ldots, \omega_N$, weights $\alpha_1, \ldots, \alpha_N$ and phase-shifts $\vartheta_1, \ldots, \vartheta_N$, such that

$$\sup_{\tau \in [0,\Delta t]} \left| u(t - \tau) - \sum_{j=1}^N \alpha_j \mathcal{L}_t u(\omega_j) \sin(\omega_j \tau - \vartheta_j) \right| \le \frac{\epsilon}{2}, \quad \forall t \in [0,T], \ \forall u \in K, \qquad \text{(B.4)}$$

where any $u \in K$ is extended by zero to negative times. It follows from Lemma 3.7, that there exists a coupled oscillator network,

$$\ddot{y} = \sigma(w \odot y + Vu + b), \qquad y(0) = \dot{y}(0) = 0,$$

with dimension $m = pN$, and $w \in \mathbb{R}^m$, $V \in \mathbb{R}^{m \times p}$, and a linear output layer $y \mapsto \widetilde{A}y$, $\widetilde{A} \in \mathbb{R}^{m \times m}$, such that $[\widetilde{A}y(t)]_j \approx \mathcal{L}_t u(\omega_j)$ for $j = 1, \ldots, N$; more precisely, such that

$$\sup_{t \in [0,T]} \sum_{j=1}^N |\alpha_j| \left| \mathcal{L}_t u(\omega_j) - [\widetilde{A}y]_j(t) \right| \le \frac{\epsilon}{2}, \quad \forall u \in K. \qquad \text{(B.5)}$$

Composing with another linear layer $B : \mathbb{R}^m \simeq \mathbb{R}^{p \times N} \to \mathbb{R}^p$, which maps $\boldsymbol{\beta} = [\beta_1, \ldots, \beta_N]$ to

$$B\boldsymbol{\beta} := \sum_{j=1}^N \alpha_j \beta_j \sin(\omega_j \Delta t - \vartheta_j) \in \mathbb{R}^p,$$

we define $A := B\widetilde{A}$, and observe that from (B.4) and (B.5):

$$\sup_{t\in[0,T]} |u(t-\Delta t) - Ay(t)| \leq \sup_{t\in[0,T]} \left| u(t-\Delta t) - \sum_{j=1}^{N} \alpha_j \mathcal{L}_t u(\omega_j) \sin(\omega_j \Delta t - \vartheta_j) \right|$$

$$+ \sup_{t\in[0,T]} \sum_{j=1}^{N} |\alpha_j| \left| \mathcal{L}_t u(\omega_j) - [\widetilde{A}y]_j(t) \right| |\sin(\omega_j \Delta t - \vartheta_j)|$$

$$\leq \epsilon.$$

$\square$

## B.5   Proof of Lemma 3.9

*Proof.* Fix $\Sigma, \Lambda, \gamma$ as in the statement of the lemma. Our goal is to approximate $u \mapsto \Sigma\sigma(\Lambda u + \gamma)$.

**Step 1: (nonlinear layer)** We consider a first layer for a hidden state $y = [y_1, y_2]^T \in \mathbb{R}^{p+p}$, given by

$$\begin{cases} \ddot{y}_1(t) = \sigma(\Lambda u(t) + \gamma) \\ \ddot{y}_2(t) = \sigma(\gamma) \end{cases}, \quad y(0) = \dot{y}(0) = 0.$$

This layer evidently does not approximate $\sigma(\Lambda u(t) + \gamma)$; however, it does encode this value in the second derivative of the hidden variable $y_1$. The main objective of the following analysis is to approximately compute $\ddot{y}_1(t)$ through a suitably defined additional layer.

**Step 2: (Second-derivative layer)** To obtain an approximation of $\sigma(\Lambda u(t) + \gamma)$, we first note that the solution operator

$$\mathcal{S} : u(t) \mapsto \eta(t), \quad \text{where } \ddot{\eta}(t) = \sigma(\Lambda u(t) + \gamma) - \sigma(\gamma), \quad \eta(0) = \dot{\eta}(0) = 0,$$

defines a continuous mapping $\mathcal{S} : C_0([0,T]; \mathbb{R}^p) \to C_0^2([0,T]; \mathbb{R}^p)$, with $\eta(0) = \dot{\eta}(0) = \ddot{\eta}(0) = 0$. Note that $\eta$ is very closely related to $y_1$. The fact that $\ddot{\eta} = 0$ is important to us, because it allows us to *smoothly* extend $\eta$ to negative times by setting $\eta(t) := 0$ for $t < 0$ (which would not be true for $y_1(t)$). The resulting extension defines a compactly supported function $\eta : (-\infty, 0] \to \mathbb{R}^p$, with $\eta \in C^2((-\infty, T]; \mathbb{R}^p)$. Furthermore, by continuity of the operator $\mathcal{S}$, the image $\mathcal{S}(K)$ of the compact set $K$ under $\mathcal{S}$ is compact in $C^2((-\infty, T]; \mathbb{R}^p)$. From this, it follows that for small $\Delta t > 0$, the second-order backward finite difference formula converges,

$$\sup_{t\in[0,T]} \left| \frac{\eta(t) - 2\eta(t-\Delta t) + \eta(t-2\Delta t)}{\Delta t^2} - \ddot{\eta}(t) \right| = o_{\Delta t \to 0}(1), \quad \forall \eta = \mathcal{S}(u), u \in K,$$

where the bound on the right-hand side is uniform in $u \in K$, due to equicontinuity of $\{\ddot{\eta} \mid \eta = \mathcal{S}(u), u \in K\}$. In particular, the second derivative of $\eta$ can be approximated through *linear combinations of time-delays of $\eta$*. We can now choose $\Delta t > 0$ sufficiently small so that

$$\sup_{t\in[0,T]} \left| \frac{\eta(t) - 2\eta(t-\Delta t) + \eta(t-2\Delta t)}{\Delta t^2} - \ddot{\eta}(t) \right| \leq \frac{\epsilon}{2\|\Sigma\|}, \quad \forall y = \mathcal{S}(u), u \in K,$$

where $\|\Sigma\|$ denotes the operator norm of the matrix $\Sigma$. By Lemma 3.8, applied to the input set $\widetilde{K} = \mathcal{S}(K) \subset C_0([0,T]; \mathbb{R}^p)$, there exists a coupled oscillator

$$\ddot{z}(t) = \sigma(w \odot z(t) + V\eta(t) + b), \quad z(0) = \dot{z}(0) = 0, \tag{B.6}$$

and a linear output layer $z \mapsto \widetilde{A}z$, such that

$$\sup_{t\in[0,T]} \left| [\eta(t) - 2\eta(t-\Delta t) + \eta(t-2\Delta t)] - \widetilde{A}z(t) \right| \leq \frac{\epsilon \Delta t^2}{2\|\Sigma\|}, \quad \forall \eta = \mathcal{S}(u), u \in K.$$

Indeed, Lemma 3.8 shows that time-delays of any given input signal can be approximated with any desired accuracy, and $\eta(t) - 2\eta(t-\Delta) - \eta(t-2\Delta)$ is simply a linear combination of time-delays of the input signal $\eta$ in (B.6).

To connect $\eta(t)$ back to the $y(t) = [y_1(t), y_2(t)]^T$ constructed in Step 1, we note that

$$\ddot{\eta} = \sigma(Au(t) + b) - \sigma(b) = \ddot{y}_1 - \ddot{y}_2,$$

and hence, taking into account the initial values, we must have $\eta \equiv y_1 - y_2$ by ODE uniqueness. In particular, upon defining a matrix $\widetilde{V}$ such that $\widetilde{V}y := V y_1 - V y_2 \equiv V\eta$, we can equivalently write (B.6) in the form,

$$\ddot{z}(t) = \sigma(w \odot z(t) + \widetilde{V}y(t) + b), \quad z(0) = \dot{z}(0) = 0. \tag{B.7}$$

**Step 3: (Conclusion)**

Composing the layers from Step 1 and 2, we obtain a coupled oscillator

$$\ddot{y}^\ell = \sigma(w^\ell \odot y^\ell + V^\ell y^{\ell-1} + b^\ell), \quad (\ell = 1, 2),$$

initialized at rest, with $y^1 = y$, $y^2 = z$, such that for $A := \Sigma\widetilde{A}$ and $c := \Sigma\sigma(\gamma)$, we obtain

$$\sup_{t \in [0,T]} \left| [Ay^2(t) + c] - \Sigma\sigma(\Lambda u(t) + \gamma) \right| \leq \|\Sigma\| \sup_{t \in [0,T]} \left| \widetilde{A}z(t) - [\sigma(\Lambda u(t) + \gamma) - \sigma(\gamma)] \right|$$

$$= \|\Sigma\| \sup_{t \in [0,T]} \left| \widetilde{A}z(t) - \ddot{\eta}(t) \right|$$

$$\leq \|\Sigma\| \sup_{t \in [0,T]} \left| \widetilde{A}z(t) - \frac{\eta(t) - 2\eta(t - \Delta t) + \eta(t - 2\Delta t)}{\Delta t^2} \right|$$

$$+ \|\Sigma\| \sup_{t \in [0,T]} \left| \frac{\eta(t) - 2\eta(t - \Delta t) + \eta(t - 2\Delta t)}{\Delta t^2} - \ddot{\eta}(t) \right|$$

$$\leq \frac{\epsilon}{2} + \frac{\epsilon}{2} = \epsilon.$$

This concludes the proof. $\qquad\qquad\qquad\qquad\qquad\qquad\qquad\qquad\qquad\qquad\qquad\qquad\qquad\square$

## B.6 Proof of Theorem 3.1

*Proof.* **Step 1:** By the Fundamental Lemma 3.6, there exist $N$, a continuous mapping $\Psi$, and frequencies $\omega_1, \ldots, \omega_N$, such that

$$|\Phi(u)(t) - \Psi(\mathcal{L}_t u(\omega_1), \ldots, \mathcal{L}_t u(\omega_N); t^2/4)| \leq \epsilon,$$

for all $u \in K$, and $t \in [0, T]$. Let $M$ be a constant such that

$$|\mathcal{L}_t u(\omega_1)|, \ldots, |\mathcal{L}_t u(\omega_N)|, \frac{t^2}{4} \leq M,$$

for all $u \in K$ and $t \in [0, T]$. By the universal approximation theorem for ordinary neural networks, there exist weight matrices $\Sigma$, $\Lambda$ and bias $\gamma$, such that

$$|\Psi(\beta_1, \ldots, \beta_N; t^2/4) - \Sigma\sigma(\Lambda\boldsymbol{\beta} + \gamma)| \leq \epsilon, \quad \boldsymbol{\beta} := [\beta_1, \ldots, \beta_N; t^2/4]^T,$$

holds for all $t \in [0, T]$, $|\beta_1|, \ldots, |\beta_N| \leq M$.

**Step 2:** Fix $\epsilon_1 \leq 1$ sufficiently small, such that also $\|\Sigma\|\|\Lambda\|\text{Lip}(\sigma)\epsilon_1 \leq \epsilon$, where $\text{Lip}(\sigma) := \sup_{|\xi| \leq \|\Lambda\|M + |\gamma| + 1} |\sigma'(\xi)|$ denotes an upper bound on the Lipschitz constant of the activation function over the relevant range of input values. It follows from Lemma 3.7, that there exists an oscillator network,

$$\ddot{y}^1 = \sigma(w^1 \odot y^1 + V^1 u + b^1), \quad y^1(0) = \dot{y}^1(0) = 0, \tag{B.8}$$

of depth 1, such that

$$\sup_{t \in [0,T]} |[\mathcal{L}_t u(\omega_1), \ldots, \mathcal{L}_t u(\omega_N); t^2/4]^T - A^1 y^1(t)| \leq \epsilon_1,$$

for all $u \in K$.

**Step 3:** Finally, by Lemma 3.9, there exists an oscillator network,

$$\ddot{y}^2 = \sigma(w^2 \odot y^2 + V^2 y^1 + b^1),$$

of depth 2, such that

$$\sup_{t \in [0,T]} |A^2 y^2(t) - \Sigma\sigma(\Lambda A^1 y^1(t) + \gamma)| \leq \epsilon,$$

holds for all $y^1$ belonging to the compact set $K_1 := \mathcal{S}(K) \subset C_0([0,T]; \mathbb{R}^{N+1})$, where $\mathcal{S}$ denotes the solution operator of (B.8).

**Step 4:** Thus, we have for any $u \in K$, and with short-hand $\mathcal{L}_t u(\boldsymbol{\omega}) := (\mathcal{L}_t u(\omega_1), \dots, \mathcal{L}_t u(\omega_N))$,

$$
\begin{aligned}
\left| \Phi(u)(t) - A^2 y^2(t) \right| \leq &\left| \Phi(u)(t) - \Psi(\mathcal{L}_t u(\boldsymbol{\omega}); t^2/4) \right| \\
&+ \left| \Psi(\mathcal{L}_t u(\boldsymbol{\omega}); t^2/4) - \Sigma \sigma(\Lambda[\mathcal{L}_t u(\boldsymbol{\omega}); t^2/4] + \gamma) \right| \\
&+ \left| \Sigma \sigma(\Lambda[\mathcal{L}_t u(\boldsymbol{\omega}); t^2/4] + \gamma) - \Sigma \sigma(\Lambda A^1 y^1(t) + \gamma) \right| \\
&+ \left| \Sigma \sigma(\Lambda A_1 y_1(t) + \gamma) - A^2 y^2(t) \right|.
\end{aligned}
$$

By step 1, we can estimate

$$
\left| \Phi(u)(t) - \Psi(\mathcal{L}_t u(\boldsymbol{\omega}); t^2/4) \right| \leq \epsilon, \quad \forall t \in [0,T], \ u \in K.
$$

By the choice of $\Sigma, \Lambda, \gamma$, we have

$$
\left| \Psi(\mathcal{L}_t u(\boldsymbol{\omega}); t^2/4) - \Sigma \sigma(\Lambda[\mathcal{L}_t u(\boldsymbol{\omega}); t^2/4] + \gamma) \right| \leq \epsilon, \quad \forall t \in [0,T], \ u \in K.
$$

By construction of $y^1$ in Step 2, we have

$$
\begin{aligned}
\left| \Sigma \sigma(\Lambda[\mathcal{L}_t u(\boldsymbol{\omega}); t^2/4] + \gamma) - \Sigma \sigma(\Lambda A_1 y_1(t) + \gamma) \right| & \\
\leq \|\Sigma\| \mathrm{Lip}(\sigma) \|\Lambda\| \left| [\mathcal{L}_t u(\boldsymbol{\omega}); t^2/4] - A^1 y^1(t) \right| & \\
\leq \|\Sigma\| \mathrm{Lip}(\sigma) \|\Lambda\| \, \epsilon_1 & \\
\leq \epsilon, &
\end{aligned}
$$

for all $t \in [0,T]$ and $u \in K$. By construction of $y^2$ in Step 3, we have

$$
\left| \Sigma \sigma(\Lambda A^1 y^1(t) + \gamma) - A^2 y^2(t) \right| \leq \epsilon, \quad \forall t \in [0,T], \ u \in K.
$$

Thus, we conclude that

$$
|\Phi(u)(t) - A^2 y^2(t)| \leq 4\epsilon,
$$

for all $t \in [0,T]$ and $u \in K$. Since $\epsilon > 0$ was arbitrary, we conclude that for any causal and continuous operator $\Phi : C_0([0,T]; \mathbb{R}^p) \to C_0([0,T]; \mathbb{R}^q)$, compact set $K \subset C_0([0,T]; \mathbb{R}^p)$ and $\epsilon > 0$, there exists a coupled oscillator of depth 3, which uniformly approximates $\Phi$ to accuracy $\epsilon$ for all $u \in K$. This completes the proof. $\quad\square$

