# OpenReview forum: "Neural Oscillators are Universal"
_NeurIPS.cc/2023/Conference — NeurIPS 2023 poster_

### Official Review · Reviewer_vPvT · 2023-07-03

**Soundness:** 3 good
**Presentation:** 3 good
**Contribution:** 3 good
**Rating:** 6
**Confidence:** 3

**Summary:**

Various machine learning architectures inspired by oscillators perform well in a variety of tasks. The authors study the approximation ability of neural networks based on oscillators. The main contribution of this paper is the construction of a neural oscillator framework combining nonlinear dynamics and a linear read-out, and the proof of a Universality theorem for neural oscillators.

**Strengths:**

* The universal approximation property is an important theoretical support for neural network architectures. This is the novel theoretical result to get a Universality theorem for neural oscillators.
* The proof strategy of the paper is relatively clear, and the illustration in Figure 1 makes it easier to understand.

**Weaknesses:**

It would be better if the authors could discuss how this universality result contributes to existing work in more detail.

**Questions:**

As a further reflection on universal approximation theory, I'm curious about the effect of network width and depth on expressiveness. Specifically, is there a minimum width that can achieve a universal approximation in infinitely deep networks or in fixed-depth networks?

The title is confusing and needs to be revised: 'universal' for what? universal approximator?

---

> ### Author Rebuttal · Authors · 2023-08-06
>
> We start by thanking the reviewer for your appreciation of the merits of our paper and your welcome suggestions to improve it. Below, we address the concerns raised by the reviewer and thank the reviewer in advance for their patience in reading our detailed reply.
>
> 1. Our main motivation for studying the universal approximation of neural oscillators is the empirical work on CoRNN and UniCoRNN, but also related systems discussed in the introduction. A priori, it is unclear what types of operators can be approximated, even in principle, with such oscillator-based architectures. Available empirical results suggest that this class includes several operators of interest, but the approach has remained without any solid theoretical foundation. Our result shows rigorously for the first time that neural oscillators are in fact very expressive, capable of approximating almost arbitrary causal operators.
>
> 2. Regarding the reviewer's question about the effect of network width and depth on expressiveness, our proof shows that a neural oscillator of fixed depth 3 and a sufficiently large width can approximate any causal operator. Upon closer inspection of the Fundamental Lemma (Lemma 3.5), we can further relate the required width of the neural oscillator to two fundamental factors:
>
> The first determining factor is the number frequencies of the sine-transform that is required to approximately reconstruct, or at least adequately distinguish, the input signals. The width of the first layer in our construction is directly proportional to this number of required frequencies. This number of required frequencies could for example be estimated in terms of the smoothness of the input signal (with smoother input signals requiring fewer frequencies). Such analysis leads to a width proportional to $width\sim N \sim \epsilon^{-1/s}$, where $s$ is the degree of smoothness of input signals (e.g. $s=1$ for continuously differentiable inputs) and $N$ the number of frequencies $\omega_1,\dots, \omega_N$.
>
> The second determining factor is the width of a shallow neural network that is required to approximate the function $\Psi$ of the Fundamental Lemma. The width of the second and third layers in our construction are directly related to the width of this neural network. Quantitative bounds for the required width can be derived in terms of e.g. the smoothness of the function $\Psi$; in general, this approach leads to an estimate of the form $width \sim \epsilon^{-N/k}$, where $k$ is the smoothness of $\Psi$ (e.g. $\Psi \in C^k$), and where we recall that $\Psi$ depends on $N$ inputs. While this gives an upper bound, it should be kept in mind that typical operators $\Phi$ (and as a consequence also the function $\Psi$) often possess additional structure, which goes beyond smoothness. Such structure can be exploited by neural networks, making the afore-mentioned upper bounds in terms of smoothness often quite pessimistic.
>
> To summarize the above discussion, the required width of the neural oscillator is related to (i) the complexity of the input functions, and (ii) the properties of the operator (reflected by the function $\Psi$), such as its smoothness. We plan to address this issue in a CRV, if accepted, and thank the reviewer for pointing out this avenue for possible extension of our work
>
> 3. We thank the reviewer for pointing out the possibly confusing title. We plan to address this in accordance with their suggestion in a CRV, if accepted.
>
> We sincerely hope that we have addressed the concerns of the reviewer to your satisfaction and kindly request you to update your assessment accordingly.

---

> > ### Comment · Reviewer_vPvT · 2023-08-15
> >
> > Thank you for addressing some of my comments. I am maintaining my score.

---

> > > ### Author Response · Authors · 2023-08-15
> > > **Thanking the reviewer**
> > >
> > > We thank the reviewer for your reply and we are at your disposal during the discussion period if you have any further
> > > questions.

---

### Official Review · Reviewer_QtjX · 2023-07-05

**Soundness:** 2 fair
**Presentation:** 3 good
**Contribution:** 2 fair
**Rating:** 6
**Confidence:** 3

**Summary:**

This paper introduces an abstract framework of neural oscillators and proves the universal approximation theorem of multi-layer neural oscillators, which shows that neural oscillators with a specific structure have the capability of approximating causal and continuous maps between spaces of continuous functions.


**Strengths:**

1. This paper provides a novel proof approach to the approximation theory of neural oscillators. They cleverly used the time-windowed sine transform and the approximation properties of traditional neural networks as a springboard to realize the proof of the entire approximation theory.
2. They explicitly construct a three-layer neural oscillator structure which is ingenious and corresponds step-by-step to their proof process.
3. The logic of the outline of the proof is clear and easy to follow.


**Weaknesses:**

The proof process is novel, and the structure is specially designed, but in terms of applicability, this design increases the limitation of the approximation theory. It is designed to approximate different items at different layers, not just based on the last loss function or the final operators to be approximated, so it does not match the gradient-based parameter training method well. For example, the first layer is supposed to approximate the time-windowed sine transform, but based on conventional gradient algorithms, this particular limitation cannot be guaranteed. In fact, it is unknown which training method can be used to achieve such a special structure with the approximation property.


**Questions:**

Can it be stated whether it is possible to train parameters by approximating such a hierarchical separate approximation with existing algorithms? Or at the application aspect, can a new training method be designed to match the structure designed in this paper?

**Limitations:**

The authors have adequately addressed the limitations.

---

> ### Author Rebuttal · Authors · 2023-08-06
>
> We start by thanking the reviewer for reading our paper and for your comments. Below, we address the concerns raised by the reviewer and thank the reviewer in advance for their patience in reading our detailed reply.
>
> 1. The main contribution of this paper is fundamentally an approximation theoretic result, which shows that the neural oscillator architecture is sufficiently expressive to approximate a very general class of causal operators. While we uncover specific mechanisms that the architecture can exploit in principle, no claim can be made that a neural oscillator, trained in practice, will take on the specific form of the neural oscillator constructed in our proof. Nor would this necessarily be desirable, as the practical optimization may discover a more efficient, yet very data- and application-specific, approximation of a given operator of interest. This stands in contrast to the specific construction of our proof which is, in some sense, data agnostic, i.e, the same structure of the model is realized whatever the underlying task is. Thus, our theoretical contribution should be viewed as an analogue of well-known, classical universal approximation theorems for e.g. multi-layer perceptrons, convolutional neural networks, recurrent neural networks, transformers etc. Such results demonstrate that these architectures are sufficiently expressive to approximate a very wide class of functions, but it is generally not expected to retrieve the particular architecture, constructed in the proof of this universal approximation theorem, in practice. Moreover, the universal approximation theory also says that the global minimum of the optimization error can be made as small as possible assuming sufficiently large model size and sufficient data samples.
>
> However, as alluded to in our paper, our proof of universality does immediately imply that an architecture combining linear dynamics with a non-linear read-out is also universal. It would be interesting to further investigate this type of architecture, which directly reflects the basic mechanisms embedded in our proof. As already argued in lines 313-318 of this paper, this structure is reminiscent of the heavily used structured state space models (Refs [8,9] in our paper) for which no universal approximation theorems exist. So, a natural next step would be to investigate if our proof also applies to these architectures which use the structure explicitly in the model itself and will be a topic for future work.
>
> We sincerely hope to have addressed your concerns, particularly the contributions of our paper in terms of universal approximation, and would kindly request the reviewer to update your assessment accordingly.

---

> > ### Comment · Reviewer_QtjX · 2023-08-12
> >
> > Thank you for your reply and your outlook on future research, which has solved my doubts to a certain extent. I think this is indeed a meaningful work, and I have improved my score accordingly.

---

> > > ### Author Response · Authors · 2023-08-14
> > > **Thanking the Reviewer**
> > >
> > > We thank the reviewer for their comments as well as for increasing our score.

---

### Official Review · Reviewer_y7S4 · 2023-07-07

**Soundness:** 3 good
**Presentation:** 4 excellent
**Contribution:** 3 good
**Rating:** 7
**Confidence:** 3

**Summary:**

The authors present a universal approximation theorem for a system of 2nd order ODEs they call "neural oscillators." At the heart of the construction is the fact that a forced harmonic oscillator with input computes a windowed sine transform of the input, up to a constant (Lemma 3.4). The other major piece (Lemma 3.5) states that windowed sine transforms taken at a finite number frequencies can be input to a nonlinear function to approximate a causal, continuous operator to desired accuracy.

**Strengths:**

1. The recent applications of nonlinear oscillators in machine learning makes the work relevant and interesting.
2. The work seems to be technically sound.
3.The discussion brings up many interesting points including feedforward structure in physical systems, recent work in polynomial memory, expressibility versus learnability.
4. The main paper and proofs in the appendix are all detailed, readable, and well-organized.


**Weaknesses:**

The paper would be helped by a more detailed discussion of where nonlinearity is needed to achieve the universal approximation result. For example, it is known that the time-delay result in Lemma 3.7 can be achieved with a linear system. See [1], for example. Then is it the case that nonlinear oscillators are only needed in the two-layer neural network construction (Lemma 3.8)? Another example would be the windowed sine transform, which can be computed by a linear system.

[1] Gu et al. (2022) "How to train your hippo: State space models with generalized orthogonal basis projections"

**Questions:**

See "weaknesses." I remain fuzzy on exactly where nonlinear dynamics are needed in the construction.

l144: Are the masses \mu being ordered here? If so, then the quality of the upper diagonal approximation will depend on the differences between the masses and the gap bound 1/epsilon should be stated explicitly for clarity.

Minor comments:

Eq 2.3: $w^L I$ -> $\text{diag}(w^L)$

l95: causally independent, not statistically independent due to common input.

Paragraph starting on l123: A sentence more of clarification here would be helpful.

l131: this would be more readable if F were replaced with f to follow the lower-case vector/upper-case matrix convention.

l140-141: It should be made more clear that we are supposing this particular structure for the symmetric matrix C.

Lemma 3.5: $t$ is not qualified. Should it be: $\forall \epsilon \exists N$, frequencies $\omega_1, \dots, \omega_N$ and a continuous mapping $\Psi$ such that $\forall t \in [0,T^2/4]$ ?


**Limitations:**

Yes.

---

> ### Author Rebuttal · Authors · 2023-08-06
>
> We start by thanking the reviewer for your appreciation of the merits of our paper and your welcome suggestions to improve it. Below, we address the concerns raised by the reviewer and thank the reviewer in advance for reading our detailed reply.
>
> 1. Regarding the reviewer's question about more detailed discussion of where nonlinearity is needed, we agree with their assessment that nonlinearity is only needed in Lemma 3.8. As rightly pointed out by the reviewer, linear oscillators are sufficient to compute the sine-transform and the time-delay operator (we also thank the reviewer for pointing out the relevant reference Gu et. al., which we will include in the CRV, if accepted).  Due to the underlying oscillatory dynamics, it is far from obvious that the output of a single hidden-layer perceptron can be reproduced by neural oscillators. The result of Lemma 3.8, which is visually represented by the bottom half of Figure 1, guarantees this. We thank the reviewer for their suggestions, and we plan to highlight the important role of Lemma 3.8 in the CRV, if accepted, and in particular, we will make it explicit that the other parts of our proof of universality do not require non-linearity.
>
> 2. Regarding the question on the mass gap: Yes, indeed the intent is to introduce an ordering of the masses, such that $\mu^\ell / \mu^{\ell-1} \sim \epsilon$ is small. We will state this, and the explicit dependency of the quality of the upper diagonal approximation on this mass gap, in a CRV, if accepted. We thank the reviewer for suggesting this clarification.
>
> 3. We thank the reviewer also for the minor comments, which we will readily incorporate in a CRV, if accepted.
>
> We sincerely hope to have addressed your concerns, particularly about delineating the specific role of the nonlinearity in our universal approximation theorem, and would kindly request the reviewer to update your assessment accordingly.

---

> > ### Comment · Reviewer_y7S4 · 2023-08-16
> >
> > I thank the authors for their reply and clarification of where nonlinear oscillators are needed. I have consequently raised my scores for confidence and rating.

---

> > > ### Author Response · Authors · 2023-08-16
> > > **Thanking the Reviewer**
> > >
> > > We thank the reviewer for appreciating our paper and our rebuttal and for raising the score. We are really grateful.

---

### Official Review · Reviewer_Ad2m · 2023-07-07

**Soundness:** 3 good
**Presentation:** 4 excellent
**Contribution:** 4 excellent
**Rating:** 7
**Confidence:** 4

**Summary:**

This work presents a comprehensive review of diverse learning architectures based on ordinary differential equations (ODEs) and explores their relationship with coupled oscillator systems. Furthermore, the paper provides a proof showcasing the universality of the coupled oscillator architecture. The universality of ODEs is a discernible outcome, as discrete dynamics of neural networks can be effectively mapped to discrete solutions of ODEs utilizing various integration methods. Consequently, the novelty of this work might be slightly diminished by this intuitive understanding. However, the significance of this research lies in its ability to formally define the problem and rigorously prove its assertions. This is particularly noteworthy considering the limited attention given to analog-based learning systems in traditional machine learning and the growing relevance of analog systems in both classical and quantum domains. The majority of the proofs presented in the paper are well-written, rigorous, and appear valid, with the exception of the proof of the Fundamental Lemma. Addressing concerns regarding the Fundamental Lemma, which is utilized to prove the main theorem, would greatly strengthen the case for accepting this work. With this consideration, I am inclined to support the acceptance of this paper.


**Strengths:**

Provides a comprehensive review of ODE-based learning architectures and their relationship to coupled oscillator systems.
Emphasizes the importance of formalizing the problem and rigorously proving the assertions, given the lack of attention to analog-based learning systems in traditional machine learning and the increasing relevance of analog systems in classical and quantum domains.

**Weaknesses:**

 The proof of the Fundamental Lemma claims the existence of a continuous function that transforms the oscillator output, which is a sine transform of an input function, into a target function. However, there seems to be a cyclic nature in the reasoning. The proof utilizes the target function, which is not known how to implement with oscillator networks composed of a reconstructing map R as the continuous function. This cyclic dependency raises questions about the validity and logic of the proof.
If one already has access to the target function, it becomes obvious that the target function can be approximated. See line 548 and the equation following it in the supplementary material as examples where this circular reasoning is apparent.

Addressing these concerns is crucial to ensure the integrity and validity of the proof presented in the paper. Providing clarification and resolving the cyclic reasoning issue would greatly enhance the readers' understanding and confidence in the proof of the Fundamental Lemma.

**Questions:**

How important is Lemma 3? The significance of Lemma 3 in the constructive proof of the mapping from neural networks to coupled oscillators warrants further examination. While its importance may not be immediately evident, providing a more detailed explanation of how Lemma 3 is utilized would greatly enhance readers' understanding of the underlying logic behind the proof. Clarifying the role and implications of Lemma 3 in the context of the proof would be beneficial for the readers' comprehension.

**Limitations:**

This work primarily concentrates on establishing a rigorous understanding of the universality of the coupled oscillator system. While this may appear somewhat self-evident to researchers engaged in developing and deploying similar systems to address real-world problems, it is important to note that the findings do not introduce any novel architectures stemming from this discovery. As a result, the impact of this work remains confined to theoretical considerations.

---

> ### Author Rebuttal · Authors · 2023-08-06
>
> We start by thanking the reviewer for your appreciation of the merits of our paper and your welcome suggestions to improve it. Below, we address the concerns raised by the reviewer and thank the reviewer in advance for their patience in reading our detailed reply.
>
>
> 1.Regarding the reviewer's concerns about the potentially cyclic nature of the Fundamental Lemma, we would like to clarify that our proof proceeds in two steps: In a first step (the fundamental lemma), we show that the underlying operator $\Phi$ can be approximated by an operator of a very specific form (given by the composition $\Psi$ with values of the sine transform). As the reviewer correctly points out, the function $\Psi$ is closely related to the operator $\Phi$; in fact, the function $\Psi$ defined there could be interpreted as a finite-dimensional analogue, or a finite-dimensional encoding, of the operator $\Phi$. The main point of the Fundamental Lemma is that we can effectively replace the operator $\Phi$ with a function $\Psi$, making the approximation by neural oscillators more tractable. This is completely analogous to classical proofs of the universal approximation of multilayer perceptrons (e.g. [Pinkus, Acta numerica, 8, 143-195]): The proof here proceeds in two steps. It is first shown that any continuous function $f(x)$ can be approximated by a specific polynomial $p(x)$, for example obtained by projecting the underlying function $f(x)$ onto the space of polynomials of a certain degree. Then, in a second step, it is shown that any such polynomial $p(x)$ can be approximated by an MLP. The first step is a reduction to a specific set of functions with additional structure. Thus, the $\Psi$ in our Fundamental Lemma is the analogue of the afore-mentioned $p(x)$ in our proof of universality for neural oscillators. We thank the reviewer for raising this point and will include a more detailed explanation along these lines in the CRV, if accepted; in particular, we will also point out the above analogy with standard universal approximation results in the theory of multilayer perceptrons.
>
> 2. Regarding the reviewer's question about Lemma 3, we presume that they refer to Lemma 3.8. This Lemma is indeed one of the most crucial ingredients in the proof of our universal approximation theorem, as it shows that the function $\Psi$ of the Fundamental Lemma can be approximated by neural oscillators. To expand on this point, we note that it is a relatively simple observation that neural oscillators can approximately compute the sine-transform (Lemma 3.6), because even linear oscillators can compute it. It is also obvious that the function $\Psi$ can be approximated by a single hidden-layer MLP, as it follows from classical universal approximation results. However, due to the underlying oscillatory dynamics, it is far from obvious that the output of a single hidden-layer neural network can be reproduced by neural oscillators. Lemma 3.8 is precisely the result that guarantees this. It is also visually represented by the bottom half of Figure 1. We thank the reviewer for this question, and will highlight the crucial role of Lemma 3.8 in the CRV, if accepted.
>
> We sincerely hope to have addressed all concerns, particularly about the Fundamental Lemma, to your satisfaction, and hope that you will update your assessment accordingly.

---

> > ### Comment · Reviewer_Ad2m · 2023-08-17
> >
> > The authors have effectively addressed my primary technical concern. I have thus adjusted my assigned scores accordingly.

---

> > > ### Author Response · Authors · 2023-08-17
> > >
> > > We thank the reviewer again for their comments as well as for increasing our score.

---

### Author Rebuttal · Authors · 2023-08-06

At the outset, we would like to thank all four reviewers for their thorough and patient reading of our article. Their criticism and constructive suggestions will enable us to improve the quality of our article. All the changes outlined below will be incorporated in a camera ready version (CRV) of the article, if accepted. We proceed to answer the points raised by each of the reviewers individually, below.

Yours sincerely,

Authors of "Neural Oscillators are Universal".

---

### Comment · Area_Chair_z8De · 2023-08-13
**Possibly a naive question**

Dear authors,

If I understand correctly, the strategy of the proof is to reduce the problem of mapping from infinite dimensional input signal to an infinite dimensional output signal to a finite to finite one. This is the fundamental lemma (Lemma 3.5) which relies on "any continuous function can be reconstructed to desired accuracy, in terms of realizations of its time-windowed sine transform (3.2) at finitely many frequencies". It seems to me that an additional assumption e.g., that the input signal is frequency bound or sufficiently smooth would be necessary. Intuitively, this approach wouldn't work, for example, for a "chirp" signal.

Please clarify if an additional assumption about smoothness is necessary and if so how it would affect the proof.

Best,
AC

---

> ### Author Response · Authors · 2023-08-14
> **Reply to AC's question**
>
> We start by thanking the Area Chair for this pertinent question. First, the AC's general intuition is spot-on. Nevertheless, we would like to reply that no additional assumption is necessary either in the Fundamental Lemma 3.5 or in the Main universal approximation Theorem 3.1, and would like to briefly explain why: The key assumption that allows us to prove Lemma 3.5 and Theorem 3.1 in this generality is the **compactness** assumption on the set of infinite-dimensional input signals $K$, which effectively replaces smoothness assumptions.
>
> As the Area Chair correctly points out, assuming a frequency bound or imposing a smoothness constraint on the input signals are indeed relevant assumptions, since both represent sufficient conditions to ensure that the input signals belong to a compact set $K$ (essentially by the Arzela-Ascoli Theorem). However, the main drawback of explicit smoothness assumptions is that they require a specific choice of a measure of smoothness (freq. bound, bound on k-th derivatives, Lipschitz bound, Holder continuity, ...); in contrast, the proposed compactness assumption is much more general, and it subsumes all of these smoothness constraints. Therefore we believe compactness to be the most general and natural assumption in this context. Moreover, such *compactness* assumptions are necessary and have a long pedigree in infinite-dimensional learning, going back to the universal approximation theorems for operators of Chen and Chen in 1995 and used more recently by Lu et al Nat. Mach. Int, 2021, Lanthaler et al Trans. Math. Appl, 2022, Kovachki et al JMLR 2020, Kovachki et al arXiv:2108.08481v3.
>
> We thank the Area Chair for pointing out this subtlety, and will add a remark in the CRV, if accepted, where we will explicitly explain how relevant smoothness constraints, such as those mentioned by the AC, imply compactness, thus making our results very widely applicable.

---

### Decision · Program_Chairs · 2023-09-21

**Decision:**

Accept (poster)

**Comment:**

This paper shows that coupled neural oscillator architecture can universally approximate mappings from continuous input signal to continuous output signal. Although the topic of the paper is relatively speculative and the theory does not tell us if such functions can be learned, reviewers have found the paper rigorous and well written. More throrough discussions on how the smoothness of input / output signals translates to required number of neurons would be helpful. I recommend to accept this paper as a poster.